



Solid Earth

# Tracing fluid transfers in subduction zones: an integrated thermodynamic and $\delta^{18}$O fractionation modelling approach

**Alice Vho**[1], **Pierre Lanari**[1], **Daniela Rubatto**[1,2], and **Jörg Hermann**[1]

[1]Institute of Geological Sciences, University of Bern, 3012 Bern, Switzerland
[2]Institut de Sciences de la Terre, University of Lausanne, 1015 Lausanne, Switzerland

**Correspondence:** Alice Vho (alice.vho@geo.unibe.ch)

Received: 12 September 2019 – Discussion started: 23 September 2019
Revised: 9 January 2020 – Accepted: 18 January 202 – Published:

**Abstract.** Oxygen isotope geochemistry is a powerful tool for investigating rocks that interacted with fluids, to assess fluid sources and quantify the conditions of fluid–rock interaction. We present an integrated modelling approach and the computer program PTLOOP that combine thermodynamic and oxygen isotope fractionation modelling for multi-rock open systems. The strategy involves a robust petrological model performing on-the-fly Gibbs energy minimizations coupled to an oxygen fractionation model for a given chemical and isotopic bulk rock composition; both models are based on internally consistent databases. This approach is applied to subduction zone metamorphism to predict the possible range of $\delta^{18}$O values for stable phases and aqueous fluids at various pressure ($P$) and temperature ($T$) conditions in the subducting slab. The modelled system is composed of a mafic oceanic crust with a sedimentary cover of known initial chemical composition and bulk $\delta^{18}$O. The evolution of mineral assemblages and $\delta^{18}$O values of each phase is calculated along a defined $P$–$T$ path for two typical compositions of basalts and sediments. In a closed system, the dehydration reactions, fluid loss and mineral fractionation produce minor to negligible variations (i.e. within 1‰) in the bulk $\delta^{18}$O values of the rocks, which are likely to remain representative of the protolith composition. In an open system, fluid–rock interaction may occur (1) in the metasediment, as a consequence of infiltration of the fluid liberated by dehydration reactions occurring in the metamorphosed mafic oceanic crust, and (2) in the metabasalt, as a consequence of infiltration of an external fluid originated by dehydration of underlying serpentinites. In each rock type, the interaction with external fluids may lead to shifts in $\delta^{18}$O up to 1 order of magnitude larger than those calculated for closed systems. Such variations can be

detected by analysing in situ oxygen isotopes in key metamorphic minerals such as garnet, white mica and quartz. The simulations show that when the water released by the slab infiltrates the forearc mantle wedge, it can cause extensive serpentinization within fractions of 1 Myr[CE1] and significant oxygen isotope variation at the interface. The approach presented here opens new perspectives for tracking fluid pathways in subduction zones, to distinguish porous from channelled fluid flows, and to determine the $P$–$T$ conditions and the extent of fluid–rock interaction.

## 1 Introduction

The subducting oceanic slab is composed of a sequence of rock types corresponding to chemical systems that undergo continuous and discontinuous reactions in response to pressure ($P$) and temperature ($T$) changes. Through its metamorphic history, the hydrated oceanic lithosphere undergoes extensive dehydration by the breakdown of low-temperature, volatile-rich minerals (e.g. Baumgartner and Valley, 2001; Baxter and Caddick, 2013; Hacker, 2008; Manning, 2004; Page et al., 2013; Poli and Schmidt, 2002). The expelled aqueous fluid migrates through the slab towards the slab–mantle interface, and it may continue rising to the mantle wedge playing a major role in triggering mass transfer and melting (Barnicoat and Cartwright, 1995; Bebout and Penniston-Dorland, 2016). Evidence for fluid circulation in subducted rocks has been extensively observed in exhumed high-pressure/ultra-high-pressure (HP/UHP)[TS1] terrains (e.g. Zack and John, 2007; Baxter and Caddick, 2013; Martin et al., 2014; Rubatto and Angiboust, 2015; Engi et

**Published by Copernicus Publications on behalf of the European Geosciences Union.**

al., 2018), but a direct link to the primary source production is often missing and the main source remains matter of debate. The characterization of fluid pathways in subduction zones has been addressed by using a variety of methods (i.e. seismicity, thermodynamic modelling, fluid inclusions, HP veins, trace element and stable isotope studies on metamorphic minerals) (e.g. Baxter and Caddick, 2013; Hacker, 2008; Hernández-Uribe and Palin, 2019; Scambelluri and Philippot, 2001; Spandler and Hermann, 2005). In particular, oxygen isotope composition of metamorphic minerals from exhumed HP rocks sheds light on the nature of the fluid reacting with those systems during metamorphism. Thus, oxygen isotope studies of HP rocks have the potential to make important contributions to the investigation of fluid sources and pathways in subduction zones (e.g. O'Neil and Taylor, 1967; Muehlenbachs and Clayton, 1972; Baumgartner and Valley, 2001; Page et al., 2013; Martin et al., 2014; White and Klein, 2014; Hoefs, 2015; Rubatto and Angiboust, 2015).

The modelling of oxygen isotopic fractionation has been traditionally addressed as an equilibrium calculation between individual mineral couples. An alternative approach follows what has been extensively adopted in the last decades for thermodynamic modelling (see reviews by Lanari and Duesterhoeft, 2019; Powell and Holland, 2008; Spear et al., 2017) and considers an evolving mineral assemblage. A pioneer model proposed by Kohn (1993) can be applied to single and closed chemical systems, i.e. for which no infiltration of external fluids in isotopic disequilibrium is allowed. Such an approach can simulate how the oxygen isotopic composition changes with $P$ and $T$, but it remains too simple for subduction zone settings, where significant fluid exchange occurs between different lithologies within the subducting slab. Baumgartner and Valley (2001) proposed a model for stable isotope fluid–rock exchange based on continuum mechanics, where infiltration profiles can be calculated, but no information is provided about the different components (minerals) of the rock, as it is regarded as a continuum. The fluid / rock ($F/R$) ratios obtained with this strategy do not correspond to the physical amount of fluid but rather represent a measurement of exchange progress.

We present a new approach that combines equilibrium thermodynamics and oxygen isotope fractionation modelling applied to multi-rock systems. This modelling technique takes advantage of the increased capability of forward modelling of complex systems achieved in the last 2 decades (Lanari and Duesterhoeft, 2019). A MATLAB©-based modelling program PTLOOP has been developed to calculate oxygen isotope fractionation between stable phases from the results of Gibbs energy minimization performed by Theriak-Domino (de Capitani and Brown, 1987; de Capitani and Petrakakis, 2010) along any fixed $P$–$T$ trajectory. The oxygen isotope variation in each mineral within the evolving assemblage is tracked using an extensive and internally consistent database for oxygen isotope fractionation (Vho et al., 2020). A graphical user interface (GUI) provides the representation

of the results. The capabilities of this software solution will be discussed in detail with an example that focuses on the characterization of (1) the effect of the dehydration reactions on the bulk $\delta^{18}O$ of a rock, (2) the effect of the influx into a subducting rock of an external fluid of distinct isotopic composition and (3) the final amount and isotopic composition of the fluid leaving the multi-rock system, e.g. infiltrating an upper unit or the mantle wedge. Petrological implications of relevant computational results are also discussed.

## 2 Modelling

### 2.1 Model geometry

The subducting oceanic lithosphere is typically composed of a section of igneous oceanic crust with its sedimentary cover (mostly < 1 km) above and an ultramafic lithospheric mantle section beneath. The geometry of the model is illustrated in Fig. 1. The target column represents a simplified section of the upper part of such oceanic lithosphere. It is composed of a layer of basaltic composition (Rock 1) overlaid by a layer of sediments (Rock 2, see below for details). Two different rock columns are considered: (1) a relatively water-rich system with altered basalts and terrigenous sediments and (2) a relatively water-poor system with unaltered basalts (of mid-ocean ridge basalt (MORB) composition) and carbonate sediments. The column has a fixed section of $1\,m^2$, while the thickness of each rock unit can be set by the user. The model is conservative with respect to the mass, while the volume of each rock type changes according to fluid loss and density variation along the $P$–$T$ path. The $P$–$T$ structure of subduction zones depends on numerous variables, including the age of the incoming lithosphere and the amount of previously subducted lithosphere (e.g. Peacock, 1990). In this study, the calculation was performed following the subduction geotherm from Gerya et al. (2002) (Fig. 2) over a pressure range of 1.3–2.6 GPa, corresponding to a depth of $\sim 45$ to $\sim 85$ km to encompass the conditions of interest for the investigated processes. The modelled temperatures range from a minimum of $350\,°C$ to a maximum of $700\,°C$. The lower limit was chosen to take into account the large uncertainties in the thermodynamic databases for many common low-$T$ metamorphic minerals that lead to unsatisfactory models for phase equilibria and mineral parageneses for low-grade rocks (Frey et al., 1991; Vidal et al., 2016). The upper limit is fixed by the wet solidus of metasediments – the present model strictly applies to subsolidus conditions.

During burial and heating the different slab lithologies undergo dehydration reactions. The produced fluid escapes from the source rock and migrates upward, likely interacting with the surrounding units of different chemical and isotopic composition. The effect of an external fluid input on the $\delta^{18}O$ value of growing minerals is strongly dependent on the isotopic composition of the infiltrating fluid ($\delta^{18}O_{fluid}$)

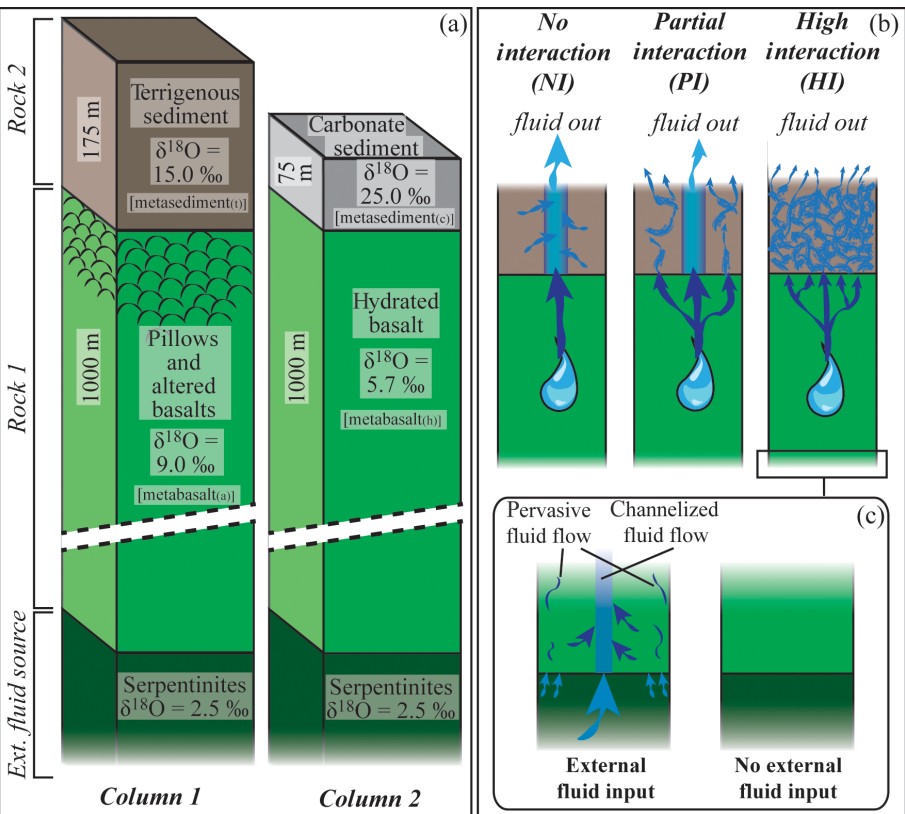

**Figure 1.** Schematic geometry of the subduction models discussed in the text. The rock column is composed of two rock types (Rock 1 and Rock 2) that can be infiltrated by an external fluid deriving from a third layer located beneath them. **(a)** Example columns used in the calculation along the $P–T$ path shown in Fig. 2 to produce the results presented in Figs. 3–7. See text for details. **(b)** Schematic representation of the three interaction cases discussed in the text. No-interaction case (NI): the fluid released by the metabasalt does not interact with the metasediment. The fluid leaving the system is a mixture of metabasalt-derived and metasediment-derived fluids. Partial-interaction case (PI): 50 % of the metabasalt-derived fluid does not interact with the metasediment, and 50 % of it equilibrates with the metasediment. The final fluid released by the system is the mixture between the unmodified metabasalt-derived fluid and the fluid deriving from the metasediment after it equilibrates with 50 % of the metabasalt-derived fluid. High-interaction case (HI): all the fluid released by the metabasalt equilibrates with the metasediment. The fluid leaving the system exits the metasediment. **(c)** Possible scenario at the base of the column. As a consequence of serpentine breakdown, serpentinite-derived fluids may infiltrate the metabasalt, exchange with it and affect the fluid infiltrating the metasediment.

and on the degree of fluid–rock interaction. The integrated $F/R$ ratio is defined here as the total mass of aqueous fluid that has passed through and interacted with the rock normalized to the mass of the rock. To explore different scenarios, three models are discussed involving different associations of fresh or altered oceanic basalts with terrigenous or carbonate sediments (Fig. 1b): (1) the interaction between the fluid released from the metamorphosed mafic crust and the overlying metasediment is negligible and the two rocks evolve independently (no-interaction case, NI); (2) part of the fluid derived from the metabasalt (50 % when not specified differently) equilibrates with the metasediment, while the other part leaves the system (partial-interaction case, PI); and (3) all the fluid released by the metamorphosed mafic crust equilibrates with the metasediment before escaping the system (high-interaction case, HI). The fluid released by the

entire system is a mixture of fluids derived from the progressive dehydration of the metabasalt and metasedimentary layers. In the NI case, both rock types behave like closed systems and the fluid is liberated from the metabasalt and from the metasediment separately. In the case of infiltration of fluid derived from the metabasalt in the metasediment, the amount of fluid released by this latter includes the fluid produced by dehydration reactions plus the excess fluid that enters the metasediment and cannot be incorporated into stable hydrous minerals.

The thickness and the degree of serpentinization of the lithospheric mantle subducting beneath the oceanic crust can be highly variable. The most important dehydration reactions in partly or fully serpentinized mantle are related to antigorite breakdown, which can release up to 12 wt % of water, playing an important role for water flows in subduction zones.

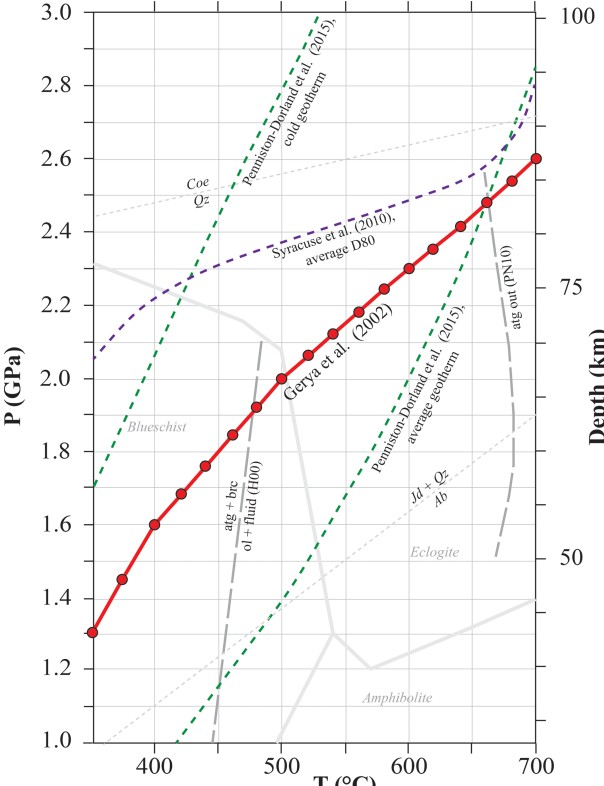

**Figure 2.** $P$–$T$ diagram showing typical oceanic subduction geotherms and the intermediate geotherm used in the calculation (red line, Gerya et al., 2002). The red spots represent the modelling steps. The average D80 geotherm from Syracuse et al. (2010) (purple dashed line), i.e. the geotherm dominated by a steep $T$ gradient at 80 km depth, which occurs at the transition from partial to full coupling, as reported by Penniston-Dorland et al. (2015) and the average slab-top geotherm from Penniston-Dorland et al. (2015) (green dashed lines) are shown for comparison. Metamorphic facies modified from Peacock (1993) and Liou et al. (2004). Serpentine breakdown reactions from Padrón-Navarta et al. (2010) (PN10) and Hermann et al. (2000) (H00). Mineral abbreviations from Whitney and Evans (2010).

Deserpentinization is assumed to result in two main subsequent fluid peaks (Padrón-Navarta et al., 2013; Scambelluri et al., 2004) related to the reactions

antigorite + brucite → olivine + chlorite + water

$\left( \sim 480\,°C, 1.7\,GPa; 1\,to\,2\,wt\,\%\,water\,released,\right.$

$\left.up\,to \sim 50\,kg\,m^{-3}\right),$ (1)

antigorite → olivine + orthopyroxene + chlorite + water

$\left( \sim 660\,°C, 2.5\,GPa; \geq 6.5\,wt\,\%\,water\,released,\right.$

$\left.\geq 170\,kg\,m^{-3}\right).$ (2)

The effect on the $\delta^{18}O$ of the metabasalts and metasediments of an external fluid influx, i.e. caused by dehydration

of the underlying serpentinites, was investigated by defining an amount of fluid with a specific $\delta^{18}O$ value that infiltrates the basaltic layer (Rock 1) at two steps of the model (480 and 660 °C along the chosen $P$–$T$ path) (Figs. 1c, 2).

## 2.2 Model strategy

The strategy behind PTLoop consists of forward modelling the evolution of the mineral assemblage and the oxygen isotope composition of a rock column composed of two lithologies (Fig. 1a) of assigned thickness and starting bulk chemical and oxygen isotope compositions along a defined $P$–$T$ path using a stepwise procedure (Fig. 2). At each $P$–$T$ step, (1) the equilibrium mineral assemblage, oxygen isotope composition of stable phases ($\delta^{18}O\,‰$ vs. Vienna Standard Mean Ocean Water, VSMOW), mass (in kg) and isotopic composition of the excess fluid for the metabasalt are calculated; (2) any fraction of excess fluid deriving from dehydration reactions in the metabasalt can be transferred to the metasediment or directly escapes the system; (3) the equilibrium mineral assemblage, $\delta^{18}O$ value of stable phases, and amount and $\delta^{18}O$ of the excess fluid for the metasediment are evaluated by accounting for the changes caused by the fluid input from the metabasalt; (4) the mass and $\delta^{18}O$ of the total fluid leaving the system are calculated. Furthermore, at each step a chosen amount of external fluid with a given $\delta^{18}O$ can be input into the metabasalt and its contribution is accounted for in the subsequent steps. This model is based on the assumption of thermodynamic equilibrium applied to a partially reactive system, whereby phases are assumed to reach chemical and isotopic equilibrium at all steps within the reactive part of the system, i.e. removing from the reactive bulk the phases that are fractionated (Lanari and Engi, 2017). Such petrological models can account for element sequestration during prograde metamorphism. Mineral fractionation in relics and fluid input or loss are two processes that are allowed to modify the reactive bulk composition. No mineral resorption is permitted. Any fluid liberated during dehydration (excess fluid) does not interact further and leaves the rock. This process is termed Rayleigh volatilization (Rumble, 1982; Valley, 1986). In natural rocks, it often occurs combined with the opposite endmember, the batch volatilization, where the produced fluid stays within the system as the mineral reaction proceeds and remains in isotopic equilibrium with the rock until the reaction is completed. In most natural cases involving oxygen isotopes, the difference between the results calculated using the two processes is negligible (Baumgartner and Valley, 2001). The released fluid is regarded as pure $H_2O$. Any other effect related to solute transport by the fluid is ignored in the calculation; the potential effect of a $CO_2$ component in the fluid is discussed in Sect. 3.2.

## 2.3 Governing equations

Equilibrium assemblage calculation for a given bulk rock composition at any $P$ and $T$ is performed with the program Theriak (de Capitani and Brown, 1987; de Capitani and Petrakakis, 2010) and is based on Gibbs energy minimization. A complete description of the minimization algorithm is given by de Capitani and Brown (1987). The reacting bulk composition may evolve in the course of the metamorphic history of a rock because of mineral fractionation, fluid loss or input of external fluids. The effective bulk composition is recalculated by PTLOOP at each subsequent stage following the strategy of Lanari et al. (2017).

As for phase assemblage determination, equilibrium is a common assumption of stable isotope transport (Baumgartner and Rumble, 1988; Baumgartner and Valley, 2001; Bowman et al., 1994; Gerdes et al., 1995a, b). Thus, a molar equilibrium constant ($K$) can be defined to describe the thermodynamic stable isotope equilibrium between two substances $i$ and $j$ (Sharp, 2017) [TS2]:

$$K = \frac{^{18}O_i / ^{16}O_i}{^{18}O_j / ^{16}O_j}. \tag{3}$$

The fractionation factor ($\alpha$) can be related to the equilibrium constant $K$ as

$$\alpha = K^{1/n}, \tag{4}$$

where $n$ is the number of exchanged atoms, normally 1 for simplicity. In isotope geochemistry, the isotopic composition is commonly expressed in terms of $\delta$ values:

$$\delta_i = \left( \frac{R_i}{R_{St}} - 1 \right) \cdot 10^3 \, (‰), \tag{5}$$

where $R_i$ and $R_{St}$ [TS3] are the isotope ratio measurements for the compound $i$ and the defined isotope ratio of a standard sample respectively. For differences in $\delta$ values or for $\delta$ values of less than $\sim 10\,‰$,

$$1000 \ln \alpha_{i-j} = \delta_i - \delta_j \tag{6}$$

is a valid approximation that is used in most cases (Hoefs, 2015; Sharp, 2017). For oxygen isotope fractionation, the equation that can reproduce most of the available calibrations describing the stable isotope fractionation function between two phases is a second-order polynomial of $10^3/T$. Hence the stable isotope fractionation between two phases $i$ (with $k$ endmembers) and $j$ (pure) as a function of $T$ is described by Eq. (7):

$$\delta^{18}O_i - \delta^{18}O_j = \sum_1^k \left( \frac{A_{k,j} \cdot 10^6}{T^2} + \frac{B_{k,j} \cdot 10^3}{T} + C_{k,j} \right)$$
$$\cdot X_{k,i} \cdot \frac{N_{k,i}}{N_i}. \tag{7}$$

The conservation of the bulk $\delta^{18}O$ in the system is described by Eq. (8):

$$\delta^{18}O_{sys} \cdot N_{sys} = \sum_{k=1}^p M_k \cdot N_k \cdot \delta^{18}O_k, \tag{8}$$

where $\delta^{18}O_i$, $\delta^{18}O_j$ and $\delta^{18}O_{sys}$ are the isotopic compositions of phase $i$, phase $j$ and the system (bulk $\delta^{18}O$) respectively; $A_{k,j}$, $B_{k,j}$ and $C_{k,j}$ are the fractionation parameters for endmember $k$ of mineral $i$ vs. phase $j$; $X_{k,i}$ is the fraction of endmember $k$ in the phase $i$; $N_{k,i}$, $N_i$ and $N_{sys}$ are the total number of moles of oxygen in endmember $k$, in mineral $i$ and in the system respectively; $p$ is the number of phases; $M_k$ is the number of moles of phase $k$; $N_k$ is the its number of oxygen; and $\delta^{18}O_k$ is its oxygen isotope composition. Given a stable mineral assemblage at any $P-T$ condition, the oxygen isotope partitioning among the stable phases is calculated by solving the linear system described by the sets of $p-1$ equatiomns of type (7) and Eq. (8); Kohn, 1993; Vho et al., 2020). In closed systems, the first term in Eq. (8) is constant. Open-system behaviour can either modify the $\delta^{18}O_{sys}$ or the number of moles of the phases ($N_{sys}$). The parameters $A$, $B$ and $C$ between phases were taken from the internally consistent database for oxygen isotope fractionation DBOXYGEN version 2.0.3 (Vho et al., 2020).

## 2.4 Starting assumptions

In order to represent the variability in the basaltic portion of the oceanic crust two different bulk compositions were used (Table 1): (1) a representative average MORB basalt (Gale et al., 2013) that has been hydrated but without any other addition or removal of element (metabasalt$_{(h)}$) and (2) a basalt that underwent extensive sea-floor alteration during hydration (Baxter and Caddick, 2013 after Staudigel et al., 1996) that will be referred to as metabasalt$_{(a)}$ in the following. Those compositions are in good agreement with other compilations reported in the literature (e.g. Sun and McDonough, 1989; Albarède, 2005; Staudigel, 2014; White and Klein, 2014). Oceanic sediments were modelled with two distinct bulk compositions (Table 1): (1) terrigenous sediment referred to as metasediment$_{(t)}$ (clay from the Mariana Trench; Hacker, 2008, after Plank and Langmuir, 1998) and (2) nano [CE2] ooze carbonate referred to as metasediment$_{(c)}$ (Plank, 2014). Nano oozes are widespread carbonate sea-floor sediments (Plank, 2014), and they are close in composition to carbonate-rich sediments observed in HP terrains (e.g. Bebout et al., 2013; Kuhn et al., 2005). Thicknesses of 1000 m for the basaltic layer, of 175 m for the clay sediment and 75 m for the carbonate sediment were chosen in order to maintain proportions between oceanic crust and sediments comparable with the values reported in various compilations (e.g. Hacker, 2008; Plank, 2014). This results in a total thickness of the rock column 2 to 3 times smaller than the real thickness to encompass the assumption of homogeneous temperature over the whole column within $\sim 20\,°C$.

To overcome the effects of possible temperature variations within the column, a discretization step size of $\sim 20\,°C$ along the $P$–$T$ path was applied.

The bulk compositions were simplified to the $Na_2O$–$CaO$–$K_2O$–$MgO$–$FeO$–$Al_2O_3$–$TiO_2$–$SiO_2$–$H_2O$ system; C is present in the initial bulk composition of the metabasalt$_{(a)}$ and the metasediment$_{(c)}$ (Table 1). MnO was excluded because it overemphasizes the stability of garnet at low metamorphic conditions ($T \leq 350\,°C$). The conditions of the garnet-in reaction in Mn-absent systems match the results obtained for garnet nucleation in natural rocks (e.g. Laurent et al., 2018). Thermodynamic modelling was performed using the internally consistent dataset of Holland and Powell (1998) and subsequent updates (tc55, distributed with Theriak-Domino 4 February 2017; see Supplement S1). The following activity models where used for the solid solutions: Holland and Powell (2003) for calcite–dolomite–magnesite; Holland and Powell (1998) for garnet, white mica and talc; Holland and Powell (1996) for omphacite; Holland et al. (1998) for chlorite; Diener et al. (2007) for amphibole. In the presented model garnet undergoes fractional crystallization both in Rock 1 and Rock 2 fractionating from the reactive bulk for the subsequent steps. The amount of initial $H_2O$ in each rock was set at saturation and is reported in Table 1. No pore fluid expulsion, diagenetic and low-grade ($T < 350\,°C$) devolatilization reactions are considered in this study (see above).

A starting bulk $\delta^{18}O$ for the metabasalt$_{(h)}$ of $5.7\%_o$ was chosen and represents the reference value for an unaltered MORB (e.g. Cartwright and Barnicoat, 1999; Eiler, 2001; Staudigel, 2014; White and Klein, 2014), while a starting bulk $\delta^{18}O$ of $9.0\%_o$ is used for metabasalt$_{(a)}$, representative of basaltic material that underwent sea-floor alteration at $T \leq 400\,°C$ (e.g. Alt et al., 1986; Cartwright and Barnicoat, 1999; Eiler, 2001; Gregory and Taylor Jr., 1981; Miller and Cartwright, 2000; Staudigel, 2014; White and Klein, 2014). The starting bulk of the terrigenous sediment of $15\%_o$ represents the average for the $\delta^{18}O$ of clastic sediments reported by Eiler (2001). The chosen $\delta^{18}O$ starting bulk of the carbonate sediment is $25\%_o$, which represents a conservative estimate of marine carbonate $\delta^{18}O$ (typically $25\%_o$–$35\%_o$; Eiler, 2001). It is $\sim 5\%_o$ higher than the values for metasedimentary carbonates in the Italian Alps (e.g. Cook-Kollars et al., 2014) that are likely to have interacted with lower-$\delta^{18}O$ fluids during subduction.

In order to define the contribution of an external fluid originating in the lithospheric mantle by serpentine breakdown, a layer of 150 m of pure serpentine containing 12 wt % bulk $H_2O$ was considered and the mass of water released at each reaction was calculated by mass balance, resulting in an input of 7800 kg of water at $480\,°C$ and of 25 350 kg at $660\,°C$ to satisfy the reactions in Eqs. (1) and (2) respectively. In order to fit the thicknesses chosen for the oceanic crust and the sedimentary layer (2 to 3 times thinner than an average lithospheric section), the 150 m of pure serpentine corre-

spond to a conservative estimate of 3000 m of serpentinized peridotite with an average serpentine content of 5 % in volume. This is in agreement with the values used by Barnes and Straub (2010) and John et al. (2011) based on the estimate by Sharp and Barnes (2004). Serpentine oxygen isotope compositions reported in the literature are highly variable (Cartwright and Barnicoat, 1999, 2003; Früh-Green et al., 2001; Mével, 2003; Miller et al., 2001), typically ranging from 1 to $10\%_o$. In mid-oceanic ridge environments, the distribution has a peak between $2\%_o$ and $5\%_o$ (Mével, 2003). A value of $2.5\%_o$ was chosen from the lower-$\delta^{18}O$ side of this peak. This results in the liberation of a fluid with a characteristic low-$\delta^{18}O$ value, but one which is still feasible for natural serpentinites; this value is clearly distinct with respect to the overlying lithologies. The effect of infiltration into the metabasalts and metasediments of serpentinite-derived fluids will become smaller as the fluid $\delta^{18}O$ becomes higher, approaching the equilibrium with the overlying lithologies. The $\delta^{18}O$ value of the released fluid is $\sim 4.5\%_o$ at $T > 550\,°C$ (serpentine or CE3 water oxygen isotope fractionation factors from Vho et al., 2020). Further details on the modelling input data are given in Supplement S2.

## 3 Results

### 3.1 Stable mineral assemblage

The evolving stable mineral assemblages and bulk water contents of each lithology, without external fluid input, were calculated for each rock composition along the prograde $P$–$T$ path. Results are provided as mode-box diagrams in Fig. 3. The $H_2O$ field represents the volume fraction of excess water in each rock type. The fluid is progressively extracted becoming isolated from the reactive part of the system. Garnet is the only phase prevented from re-equilibrating in the model, thus fractionating from the reactive bulk composition. Below $450\,°C$ and 1.80 GPa, in the metabasalts glaucophane, actinolite and lawsonite comprise $\sim 80$ vol. % of the paragenesis, with minor phengite ($Si_{apfu} = 3.67$–$3.63$, $X_{Mg} = 0.62$–$0.56$ in metabasalt$_{(h)}$; $Si_{apfu} = 3.68$–$3.65$, $X_{Mg} = 0.67$–$0.58$ in metabasalt$_{(a)}$), omphacite ($X_{Na} = 0.45$–$0.42$, $X_{Mg} = 0.81$–$0.72$), chlorite and titanite. Metabasalt$_{(h)}$ is richer in $SiO_2$, FeO and MgO with respect to the metabasalt$_{(a)}$, and chlorite is stable up to $480\,°C$. Metabasalt$_{(a)}$ contains $\sim 5$ vol. % of Ca carbonate that remains stable over the entire $P$–$T$ path. For either composition, the volume of glaucophane, actinolite and lawsonite gradually decreases from $480\,°C$ and $\sim 1.90$ GPa until complete consumption at 600–$620\,°C$ and 2.30–2.36 GPa. Those represent the major hydrous phases contributing to the dehydration, while a secondary role is played by talc and zoisite at higher $P$–$T$ conditions ($T \geq 580\,°C$, $P \geq 2.24$ GPa). Most of the water still retained in the rocks is stored in phengite, the abundance of which is primarily controlled by bulk $K_2O$ content, higher in the

**Table 1.** Bulk compositions used for the simulations.

|  | $SiO_2$ | $TiO_2$ | $Al_2O_3$ | $FeO_T$ | MnO | MgO | CaO | $Na_2O$ | $K_2O$ | Total | $H_2O$ | $CO_2$ |
|---|---|---|---|---|---|---|---|---|---|---|---|---|
| Normal MORB[1] | 50.47 | 1.68 | 14.70 | 10.43 | 0.18 | 7.58 | 11.39 | 2.79 | 0.16 | 99.38 | NA | NA |
| Metabasalt$^5_{(h)}$ | 50.47 | 1.68 | 14.70 | 10.43 | 0.00 | 7.58 | 11.39 | 2.79 | 0.16 | 99.20 | 5.17[6] | 0.00 |
| Altered basalt[2] | 43.47 | 1.06 | 14.74 | 5.98 | 0.16 | 6.32 | 12.22 | 1.96 | 0.53 | 86.44 | 7.63 | 2.80 |
| Metabasalt$^5_{(a)}$ | 43.47 | 1.06 | 14.74 | 5.98 | 0.00 | 6.32 | 12.22 | 1.96 | 0.53 | 86.28 | 5.55[6] | 2.80 |
| Terrigenous sediment[3] | 49.80 | 0.60 | 14.70 | 7.30 | 2.10 | 3.10 | 3.50 | 3.10 | 3.60 | 87.80 | 10.50 | 0.00 |
| Metasediment$^5_{(t)}$ | 49.80 | 0.60 | 14.70 | 7.30 | 0.00 | 3.10 | 3.50 | 3.10 | 3.60 | 85.70 | 4.43[6] | 0.00 |
| Carbonate sediment[4] | 32.36 | 0.40 | 8.78 | 2.91 | 0.12 | 1.45 | 23.16 | 1.96 | 1.66 | 72.80 | 8.78 | 18.20 |
| Metasediment$^5_{(c)}$ | 32.36 | 0.40 | 8.78 | 2.91 | 0.00 | 1.45 | 23.16 | 1.96 | 1.66 | 72.68 | 1.95[6] | 18.20 |

[1] Gale et al. (2013). [2] Baxter and Caddick (2013) after Staudigel et al. (1996). [3] Mariana clay from Plank and Langmuir (1998). [4] Nano ooze from Plank (2014). [5] Used for the thermodynamic modelling. [6] Water content at saturation at 350 °C and 1.3 GPa. The calculated initial water content is consistent with values from the literature at these conditions (e.g. Poli and Schmidt, 1998; Hacker et al., 2003). NA: not available.

metabasalt$_{(a)}$, and that remains stable beyond the model conditions. Garnet production starts at $\sim 500$ °C and $\sim 2.00$ GPa ($X_{alm} = 0.60$, $X_{grs} = 0.35$), and garnet grows continuously until constituting $\sim 20$ vol. % of the metabasalt$_{(a)}$ and $\sim 35$ % of the metabasalt$_{(h)}$.

In the metasediment$_{(c)}$ calcium carbonate, quartz, phengite ($Si_{apfu} = 3.42$, $X_{Mg} = 0.53$) and jadeite (jadeite content $X_{jd} = 0.95$, $X_{Na} = 0.96$, $X_{Mg} = 0.39$) compose $\sim 80$ vol. % of the solids at 350 °C and 1.30 GPa. Glaucophane and lawsonite are stable up to 460 and 560 °C respectively. Jadeite abundance increases from $\sim 10$ vol. % at 440 °C to $\sim 16$ vol. % at 460 °C; ankerite is stable in minor amounts ($\leq 3$ vol. %) at 440–560 °C. Garnet ($X_{alm} = 0.57$, $X_{grs} = 0.41$) is stable only at 540–580 °C and 2.12–2.24 GPa, reaching $\sim 5$ vol. %, and is then preserved because of the assumption of fractionation from the bulk in the model. The metasediment$_{(t)}$ at $T < 500$ °C shows a paragenesis of phengite ($Si_{apfu} = 3.68$–3.55, $X_{Mg} = 0.53$–0.44), glaucophane, lawsonite, quartz and omphacite ($X_{Na} = 0.50$–0.48, $X_{Mg} = 0.87$–0.70), with minor titanite. At 520 °C and 2.06 GPa, lawsonite is consumed and the amphibole proportion reduces from $\sim 30$ vol. % to $< 20$ vol. %, while garnet ($X_{alm} = 0.63$, $X_{grs} = 0.34$) is produced and reaches $\sim 10$ vol. %. At these conditions, the omphacite content decreases and a clinopyroxene of more jadeitic composition ($X_{jd} = 0.72$, $X_{Na} = 0.76$, $X_{Mg} = 0.45$) becomes stable. These models are in line with the first-order mineralogical changes observed in subducted (and exhumed) crustal material. Thermodynamic calculations predict the coexistence of a calcic amphibole and a sodic amphibole in the metabasalts and of jadeite and omphacite in the metasediment$_{(t)}$. From an oxygen isotope partitioning perspective, the interpretation of the modelled coexistence of a sodic and a calcic amphibole either as two endmembers of a solid solution or as coexisting minerals is equivalent. The same applies to pyroxenes, for which the modelled coexistence of two pyroxenes can be interpreted as a continuous solid solution. Therefore, this does not affect

the oxygen isotope partition model final results for the other phases and the bulk.

### 3.2 Production of aqueous fluid

At the initial conditions, all the lithologies are saturated in $H_2O$ (Table 1). Up to 500 °C, lawsonite, actinolite and glaucophane are the main repositories of $H_2O$ in the metabasalts, followed by chlorite and minor phengite. A significant pulse of water is modelled at 500–520 °C and 2.00–2.06 GPa for both the metabasalts (Fig. 3a, b). This pulse is caused by a decreasing abundance of lawsonite and amphibole and a breakdown of chlorite followed by growth of garnet and omphacite. This first dehydration stage releases $\sim 25$ % of the total water loss from the metabasalt$_{(a)}$ ($\sim 4.0$ vol. % $H_2O$ liberated) and $\sim 45$ % from the metabasalt$_{(h)}$ ($\sim 6.5$ vol. % $H_2O$ liberated). The second significant pulse in the metabasalts occurs at 620–640 °C and 2.36–2.42 GPa, releasing $\sim 40$ % of the total water loss from the metabasalt$_{(a)}$ and $\sim 15$ % from the metabasalt$_{(h)}$. This pulse is caused by the final breakdown of lawsonite and of amphibole in the metabasalt$_{(a)}$ (Fig. 3b). In the metabasalt$_{(h)}$, glaucophane and actinolite breakdown takes place at 600 °C and 2.30 GPa. Growth of talc only incorporates half of the water released by amphibole breakdown, causing an intermediate fluid pulse of minor magnitude at these $P$–$T$ conditions (Fig. 3a). If an initial undersaturated basaltic composition was considered, the amount of released fluid due to the breakdown of hydrous phases would be smaller.

The metasediment$_{(c)}$ is the rock type that dehydrates the least: the two main pulses of fluid production are at 480 °C and 1.92 GPa ($\sim 0.4$ vol. % $H_2O$ liberated) and from 540 °C and 2.12 GPa to 560 °C and 2.18 GPa ($\sim 1.7$ vol. % $H_2O$ liberated), caused by a breakdown of glaucophane and lawsonite respectively (Fig. 3c). The water produced from these two dehydration stages represents $< 0.02$ wt % of the total water released by the system composed of metabasalt$_{(h)}$ and metasediment$_{(c)}$. In the metasediment$_{(t)}$, the main fluid pulse

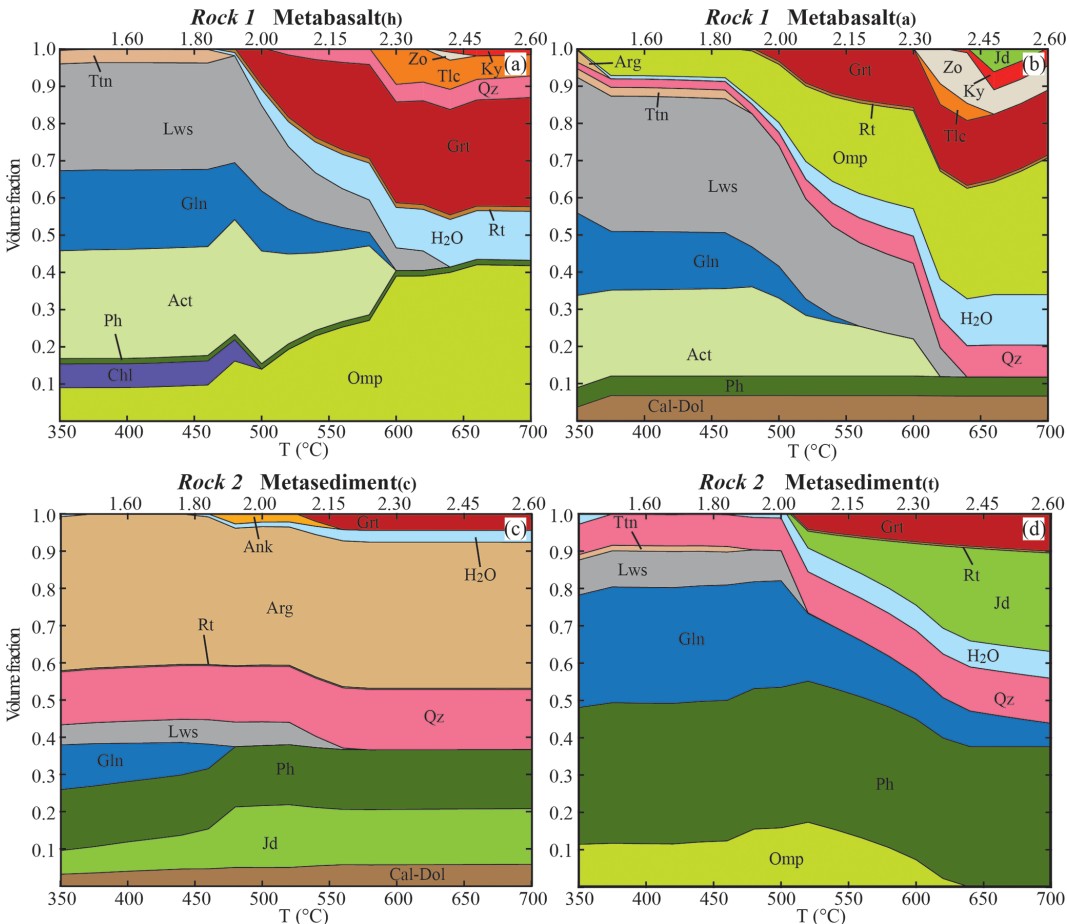

**Figure 3.** Mode-box diagrams showing the evolution of the mineral assemblages and fluid during subduction of the different rock types along the geotherm shown in Fig. 2. The initial $H_2O$ content is $< 1$ vol % in the metabasalts and in the metasediment$_{(c)}$ and $\sim 3$ vol. % in the metasediment$_{(t)}$ (see Table 1 for details). Garnet fractionation is applied to all the lithologies. The volume fraction of garnet shown at each step represents the sum of the fractionated and newly grown garnet. The phase proportions refer to the NI case, where the $H_2O$ content is the excess (-free) CE4 $H_2O$ produced by each rock type evolving independently. The excess water is fractionated at each step and the volume fraction displayed represents the sum of the fractionated and the newly produced water. Mineral abbreviations from Whitney and Evans (2010).

occurs at 520 °C and 2.06 GPa ($\sim 3.0$ vol. % of $H_2O$ liberated), caused by the breakdown of lawsonite and a decrease in glaucophane abundance (Fig. 3d). The water produced from this dehydration stage represents $\sim 0.07$ wt % of the total water released by the system composed of metabasalt$_{(a)}$ and metasediment$_{(t)}$.

As specified in Sect. 2.2, the aqueous fluid used for the calculation considers only the water component ($a_{H_2O} = 1$). The release of $CO_2$ occurs during progressive metamorphism of metabasalt$_{(a)}$ and metasediment$_{(c)}$. $CO_2$ is absent or present in negligible amounts ($< 1$ mol % of the total fluid) up to 440 °C and 1.76 GPa. At higher temperatures, the X($CO_2$) content increases significantly up to $\sim 10$ mol % in the fluid released by metabasalt$_{(a)}$ and $\sim 30$ mol % in the fluid released by metasediment$_{(c)}$ at 700 °C and 2.60 GPa. However, the total amount of fluid produced at these conditions is negligible ($< 0.01$ vol. %). The oxygen isotope fraction-

ation database used for this study (Vho et al., 2020) does not include fractionation data for $CO_2$. Limited calibrations are available for oxygen isotope fractionation between $H_2O$ and $CO_2$ (Friedman and O'Neil, 1977; O'Neil and Adami, 1969; Zheng, 1994). The fractionation values diverge significantly, for example between $-4.9\,‰$ and $-8.3\,‰$ at 450 °C and between $1.5\,‰$ and $-4.4\,‰$ at 700 °C. At 520 °C, the fluid released by metabasalt$_{(a)}$ contains 2 mol % of $CO_2$ and 6 mol % TS4–7 mol % at 620–640 °C. The consideration of $CO_2$ would produce a negligible shift on the fluid $\delta^{18}O$ ($0.1\,‰$–$0.2\,‰$ at 520 °C and fractionation used). The amount of $CO_2$ in metasediment$_{(c)}$ derived-fluid is 2 mol % at 480 °C and 5 mol % at 540 °C. The consideration of $CO_2$ would also produce negligible to minor shifts in the fluid $\delta^{18}O$ ($0.1\,‰$–$0.3\,‰$ at 480 °C and $0.2\,‰$–$1.1\,‰$ at 620 °C, depending on the fractionation).

## 3.3 Oxygen isotope composition

The largest initial bulk $\delta^{18}O$ difference occurs between metabasalt$_{(h)}$ and metasediment$_{(c)}$ (14.3‰, the relatively water-poor system), while the smallest initial bulk $\delta^{18}O$ difference is observed between metabasalt$_{(a)}$ and metasediment$_{(t)}$ (6.0‰, the relatively water-rich system). In the following, the results are presented in detail for a selection of two endmember scenarios (Figs. 4, 5): (1) metasediment$_{(c)}$ associated with metabasalt$_{(h)}$ and (2) metasediment$_{(t)}$ associated with metabasalt$_{(a)}$ when not specified differently. Other scenarios (i.e. metabasalt$_{(h)}$ associated with metasediment$_{(t)}$ and metabasalt$_{(a)}$ associated with metasediment$_{(c)}$) give intermediate results in terms of oxygen isotope composition variations as a consequence of fluid–rock interaction. Further details and the results for the intermediate scenarios are given in Supplement S3.

### 3.3.1 Bulk oxygen isotope compositions

For rocks that undergo only dehydration reactions, the starting bulk $\delta^{18}O$ evolves as a consequence of garnet and fluid fractionation (Fig. 4). The bulk $\delta^{18}O$ shift related to water fractionation is within 0.2‰, while the shift due to garnet fractionation is within 0.5‰ (Fig. 4c, d). Since water has typically heavier $\delta^{18}O$ with respect to the bulk and the garnet a lighter one, the two effects produce opposite trends. The combination of both effects results in a shift in the bulk $\delta^{18}O$ in the considered lithologies restricted to $< 0.3$‰. This in turn leads to negligible ($< 0.2$‰) variations in the $\delta^{18}O$ values of the stable phases.

In the metasediments, the progressive interaction with the fluid from the metabasalts causes a decrease in the bulk $\delta^{18}O$ (Fig. 5) that is controlled by the amount and $\delta^{18}O$ of the incoming fluid. A significant decrease starts at 480 °C, where the amount of water released by the metabasalts increases of about 1 order of magnitude from $< 0.05$ vol. % to $\sim 0.3$ vol. % due to partial consumption of amphibole and lawsonite. The maximum shift in bulk $\delta^{18}O$ was calculated for the metasediment$_{(c)}$ interacting with the metabasalt$_{(h)}$-derived fluid at $-12.9$‰ for the HI case (integrated $F/R$ ratio $= 0.75$ kg kg$^{-1}$), while it is $-8.7$‰ for the PI case (integrated $F/R$ ratio $= 0.38$). The shift in the bulk $\delta^{18}O$ of the metasediment$_{(t)}$ interacting with the metabasalt$_{(a)}$-derived fluid is $-1.5$‰ for the HI case (integrated $F/R$ ratio $= 0.35$) and $-2.7$‰ for the PI case (integrated $F/R$ ratio $= 0.18$).

### 3.3.2 Oxygen isotope composition of mineral phases

Since at infinite temperature the fractionation between any two phases approaches 0‰, a general trend in the reduction of oxygen isotope fractionation between the stable phases with increasing metamorphic grade is observed in all lithologies and is a result of the temperature increase (Fig. 4a, b, e, f). As a consequence, mineral phases typically heavier than the bulk (i.e. quartz and carbonates) become isotopically lighter with increasing metamorphic conditions, and the mineral phases typically lighter than the bulk (i.e. rutile, garnet and titanite) become isotopically heavier. Such variations are limited (i.e. within 1.0‰) for most of the phases, with the exception of quartz, calcite and rutile that may vary up to 3.0‰ in response to temperature variation only in the considered range.

In the case of ingress of the low-$\delta^{18}O$ fluid from the metabasalts in the metasedimentary rocks (PI and HI cases), the mineral $\delta^{18}O$ values decrease progressively with respect to the NI case following the trend of the bulk $\delta^{18}O$ (Fig. 5). For instance, in the case of NI the $\delta^{18}O$ of quartz in the metasediment$_{(t)}$ decreases from 19.4‰ to 17.3‰ ($-2.1$‰) and that of quartz in the metasediment$_{(c)}$ from 28.0‰ to 26.5‰ ($-1.5$‰) over the total temperature range modelled. In the PI and HI cases, the $\delta^{18}O$ shift in quartz is respectively $-3.8$‰ and $-5.0$‰ in the metasediment$_{(t)}$ and $-10.0$‰ and $-13.9$‰ in the metasediment$_{(c)}$. Hence, the final quartz $\delta^{18}O$ values (at 700 °C, 2.60 GPa) for the HI case in the metasediment$_{(t)}$ and in the metasediment$_{(c)}$ are respectively 3.0‰ and 13.0‰ isotopically lower than the expected values in the case of NI (Fig. 5). The maximum shift in $\delta^{18}O$ (i.e. between NI and HI cases) for the other stable phases is within those values. In the metasediment$_{(t)}$, the $\delta^{18}O$ values of phengite, glaucophane, jadeite, rutile and garnet decrease by 3.0‰ and those of omphacite by 2.4‰ from the NI to the HI case. Lawsonite and titanite are not stable after the first significant input of metabasalt-derived fluid (500 °C, 2.00 GPa), and their $\delta^{18}O$ values decrease by 1.1‰ and 0.3‰ respectively from the NI to the HI case. In the metasediment$_{(c)}$, the $\delta^{18}O$ values of dolomite, jadeite, phengite, rutile and aragonite decrease by a maximum of 13.1‰, of lawsonite and ankerite by a maximum of 10.3‰ and 9.5‰ respectively. Garnet crystallizes only between 540 and 580 °C and its $\delta^{18}O$ value in the HI case is 5.9‰ lower than the one in the NI case.

### 3.3.3 Oxygen isotope composition of the fractionated fluids

The $\delta^{18}O$ of the fractionated water from each rock type at each step is in isotopic equilibrium with the stable mineral assemblage at the given conditions. In the temperature range where most of the fluid is released (i.e. $T > 480$ °C), the $\delta^{18}O$ of the fluid is $7.0 \pm 0.5$‰ for the metabasalt$_{(h)}$ and $10.0 \pm 0.5$‰ for the metabasalt$_{(a)}$. At $T > 480$ °C, the water released in the NI case by the metasediment$_{(c)}$ has a $\delta^{18}O = 24.2$–25.6‰ and that released from the metasediment$_{(t)}$ has a value of 15.4–16.4‰ (Fig. 6). The $\delta^{18}O$ of the fluid leaving the system, e.g. infiltrating an upper layer or the mantle wedge, results from the mixing of the water released from the metabasalts and the overlying metasediments, and it derives from the balance between the amount of fluid released by each rock type and its $\delta^{18}O$ value (Fig. 6a, b).

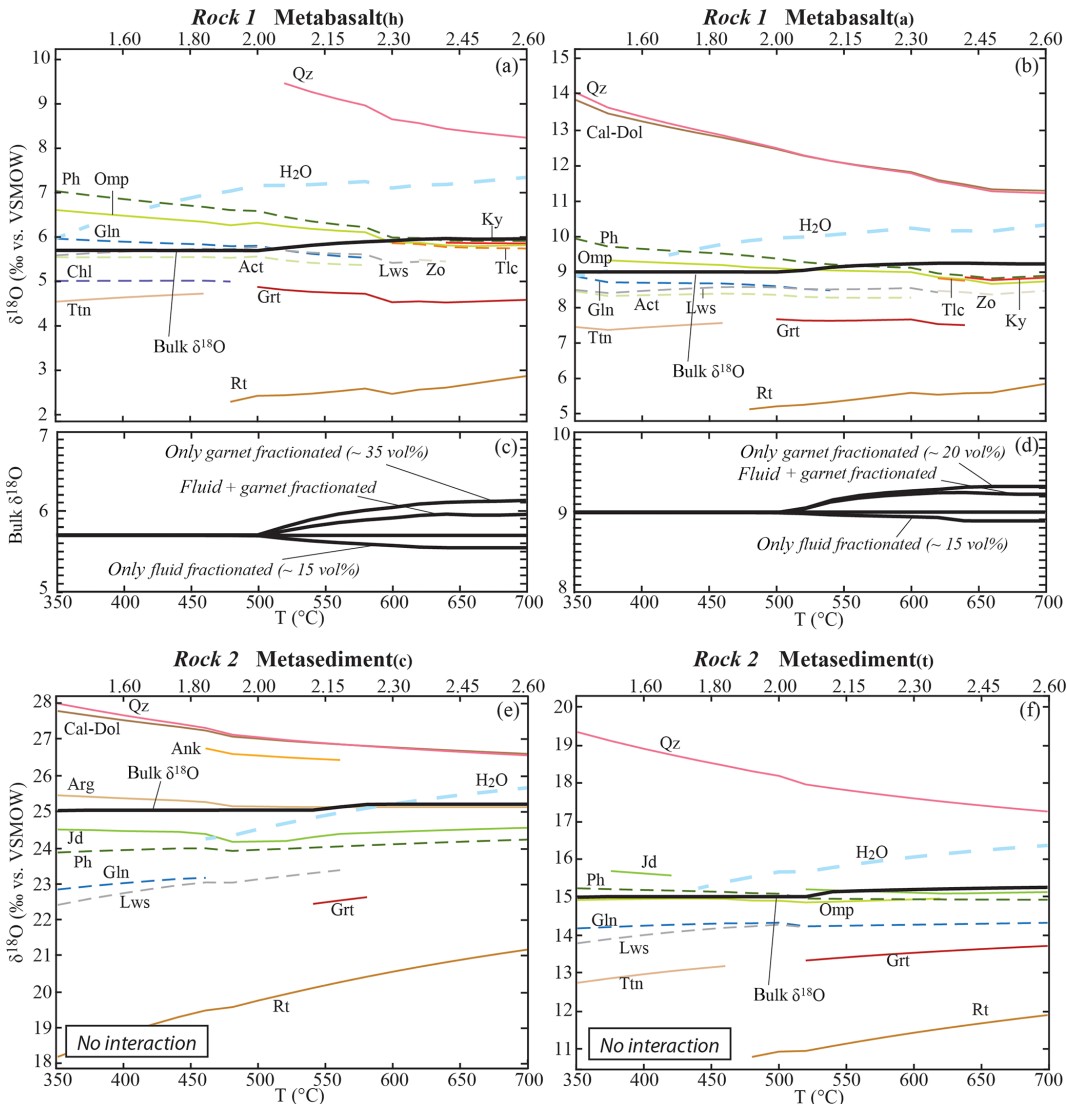

**Figure 4.** Calculated bulk and mineral $\delta^{18}O$ values along the geotherm shown in Fig. 2. Bulk $\delta^{18}O$: black solid line. Hydrous mineral $\delta^{18}O$: coloured dashed lines. Anhydrous mineral $\delta^{18}O$: coloured solid lines. Released $H_2O$: thick blue dashed lines. **(a, b)** Modelled mineral, bulk and released fluid $\delta^{18}O$ values for the metabasalt$_{(h)}$ and the metabasalt$_{(a)}$ considering garnet fractionation and excess fluid loss and in the absence of external fluid input. **(c, d)** Quantification of the effects of garnet fractionation and fluid loss on the bulk $\delta^{18}O$ of the metabasalts. **(e, f)** Modelled mineral, bulk and released fluid $\delta^{18}O$ values for the metasediment$_{(c)}$ and the metasediment$_{(t)}$ considering garnet fractionation and excess fluid loss and in the absence of external fluid input. Scale on the top $x$ axis indicates P (GPa). Mineral abbreviations from Whitney and Evans (2010).

In the NI case, the $\delta^{18}O$ of water leaving the system is up to 1‰ higher than the composition of the fluid released by the metabasalt because of the minor input from the metasediment at around 500–550 °C. The only exceptions are for the interaction between the metabasalt$_{(a)}$ and the metasediment$_{(t)}$ at $T$ of $\sim 450$ and $\sim 700$ °C. At these conditions, the $\delta^{18}O$ values of the final fluid are up to 5‰ higher than the metabasalt$_{(h)}$-derived fluid (Fig. 6b). This increase is caused by a predominance of the metasediment-derived fluid at those conditions; however, the amount of high-$\delta^{18}O$ fluid

represents $< 0.1$ wt % of the rock column and $\sim 1$ wt % of the total released fluid (Fig. 6b).

In the case of interaction between the metasediments and the metabasalt-derived fluid, part of or all this fluid reacts with the overlying metasediment before leaving the system. The final $\delta^{18}O$ of the fluid is controlled by the integrated $F/R$ ratio in the metasediments and their buffering capacity. In the HI case, the $\delta^{18}O$ of the released fluid has a dominant sedimentary signature at $T < 500$–520 °C, before the first fluid pulse from the metabasalt (14.5‰– 15.5‰ for the metasediment$_{(t)}$ and 23.0‰–24.0‰ for

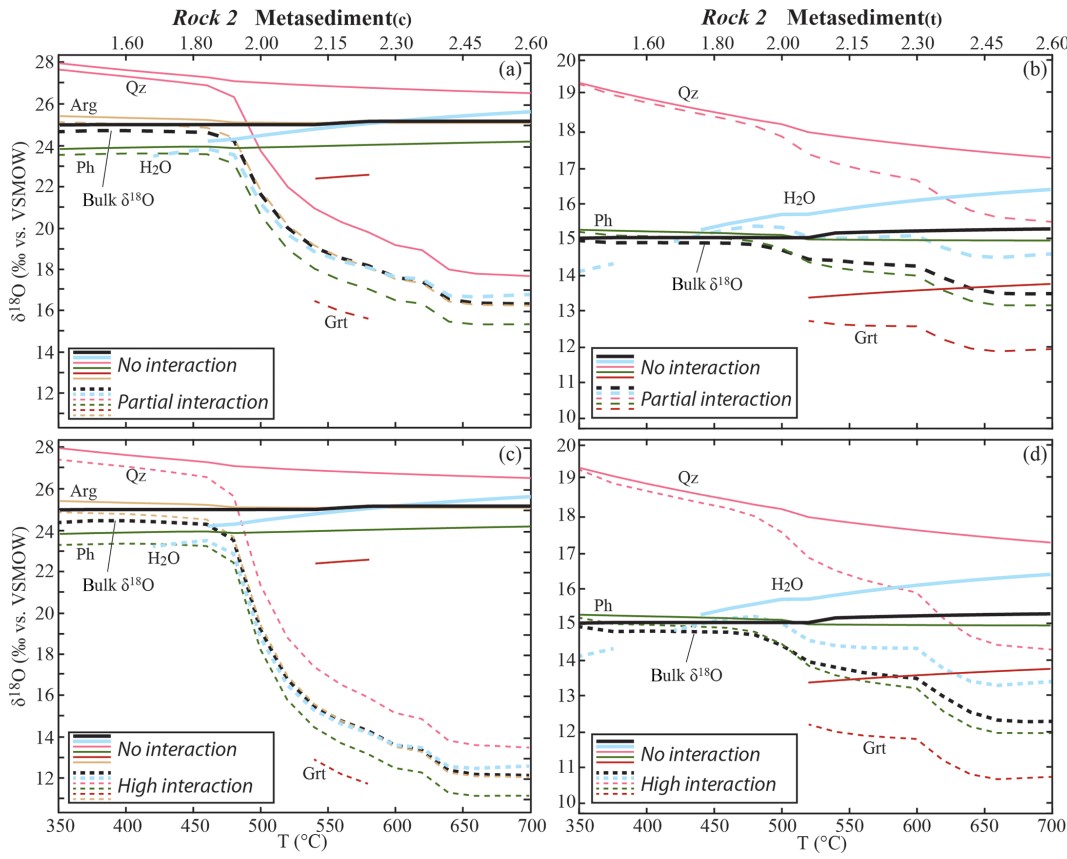

**Figure 5.** Calculated bulk and mineral $\delta^{18}O$ values in the metasediments along the geotherm shown in Fig. 2 in the case of interaction with the metabasalt-derived fluid. Only bulk, released $H_2O$ and representative mineral $\delta^{18}O$ values are shown for clarity. Garnet fractionation and excess fluid loss are considered. **(a)** Metasediment$_{(c)}$: evolution of the $\delta^{18}O$ values in the case of NI (continuous lines) and PI (dashed lines). **(b)** Metasediment$_{(t)}$: evolution of the $\delta^{18}O$ values in the case of NI (continuous lines) and PI (dashed lines). **(c)** Metasediment$_{(c)}$: evolution of the $\delta^{18}O$ values in the case of NI (continuous lines) and HI (dashed lines). **(d)** Metasediment$_{(t)}$: evolution of the $\delta^{18}O$ values in the case of NI (continuous lines) and HI (dashed lines). Scale on the top $x$ axis indicates P (GPa). Mineral abbreviations are from Whitney and Evans (2010).

the metasediment$_{(c)}$, Fig. 6). The first metabasalt-derived fluid pulse (500–520 °C, see above) causes a drop in the bulk $\delta^{18}O$ values of the total released fluids of 0.7 ‰ for the metabasalt$_{(a)}$–metasediment$_{(t)}$ association and of 6.3 ‰ for the metabasalt$_{(h)}$–metasediment$_{(c)}$ association. The second metabasalt-derived fluid pulse (620–640 °C, see above) causes a second decrease in the $\delta^{18}O$ values of the total released fluid equal to 1.0 ‰ for both the lithological associations.

### 3.3.4 Input of serpentinite-derived fluid

Ultramafic rocks tend to undergo episodic dehydration (see above). In the following, the effects caused by the input of a serpentinite-derived fluid at the base of the rock column in the case of HI are described (Fig. 7). The input of the amount of water with a $\delta^{18}O$ of 4.5 ‰ (see above), corresponding to the dehydration of 150 m of pure serpentine at 480 °C (2.0 wt % $H_2O$) and 660 °C (6.5 wt % $H_2O$), has a

limited impact on the $\delta^{18}O$ of the system CE5. It produces a decrease of < 0.2 ‰ in the final bulk $\delta^{18}O$ of the metabasalts, up to ∼ 1.0 ‰ in the metasediment$_{(c)}$ (Fig. 7a) and < 0.5 ‰ in the metasediment$_{(t)}$ with respect to the HI case with no serpentinite-derived fluid (Fig. 7b). The largest decrease occurs at 660 °C, where the second pulse of serpentinite-derived fluid enters the system, resulting in ca. −0.1 ‰, −0.3 ‰ and −1.0 ‰ for the metabasalts, metasediment$_{(t)}$ and metasediment$_{(c)}$ $\delta^{18}O$ bulk values respectively (Fig. 7). Even by increasing the thickness of the serpentinite by a factor of 2, the variations in bulk $\delta^{18}O$ values with respect to the HI case with no serpentinite-derived fluid are < 1.0 ‰ for any rock type with the exception of the metasediment$_{(c)}$, for which the bulk $\delta^{18}O$ decreases by 2.0 ‰ with respect to the HI case with no serpentinite-derived fluid (Supplement S3). The effect of the serpentinite-derived fluid on the metabasalts remains the same for any interaction case (NI, PI and HI), while the variations in the $\delta^{18}O$ of metasedimentary rocks

decreases to zero with decreasing infiltration of external fluid (Supplement S3).

## 4 Discussion

### 4.1 Effect of stable assemblage evolution and phase fractionation on the bulk $\delta^{18}O$

CE6 The changes in mineral assemblage, modes and compositions along a prograde $P$–$T$ path control (1) the oxygen isotope partitioning between the stable phases and (2) the amount and $\delta^{18}O$ of the fluid released by the system. At the same time, oxygen isotope fractionation between the stable phases is controlled by temperature. Thus, the effects of evolving paragenesis and increasing temperature are systematically overlapping in nature. In the case of a closed system, the bulk concentrations of $^{18}O$ and $^{16}O$ remain constant and a change in one phase is compensated for exactly by adjustments in other phases (Baumgartner and Valley, 2001; Kohn, 1993). In this situation, major changes in mineral assemblage do not play a significant role in shifting the $\delta^{18}O$ of stable phases: this is demonstrated by the limited ($< 0.5\%_o$) shift in $\delta^{18}O$ values of quartz, garnet, phengite, omphacite and rutile in the metabasalt$_{(h)}$ after (1) the breakdown of amphibole and lawsonite and (2) the crystallization of talc and kyanite over a narrow temperature range between 500 and 580 °C (Fig. 4a, b). Similar effects can also be anticipated for rocks with different chemical composition that undergo major changes in the mineral assemblage (see Supplement S4).

In a closed system evolving at equilibrium, the initial chemical bulk composition and bulk $\delta^{18}O$ do not change along the $P$–$T$ evolution. However, in metamorphic rocks, compositional zoning and metamorphic overgrowths are often preserved in refractory minerals (Lanari and Engi, 2017) indicating that parts of the minerals have become isolated from the reactive volume of the rock. This scenario is commonly referred to as "partially re-equilibrated open systems", because the chemical and the isotopic compositions vary as a consequence of the fractionation of solid and fluid phases (i.e. garnet fractionation and excess fluid removal) even in the absence of external fluid input. Phase fractionation is expected to affect the bulk $\delta^{18}O$ as function of both the amount of fractionated or expelled phases and their isotopic composition. Fractionation of a phase lighter than the bulk in $\delta^{18}O$ leads to an increase in the reactive bulk $\delta^{18}O$ value, while fractionation of an isotopic heavier phase leads to a decrease in the reactive bulk $\delta^{18}O$ value.

The most common example of fractionating metamorphic mineral is garnet, which systematically records compositional zonation at low- to medium-grade (Evans, 2004; Giuntoli et al., 2018; Konrad-Schmolke et al., 2008; Spear, 1988; Tracy, 1982). Therefore, garnet fractionation was incorporated into the model in order to better approximate the behaviour of natural systems. Note that this effect is reduced at

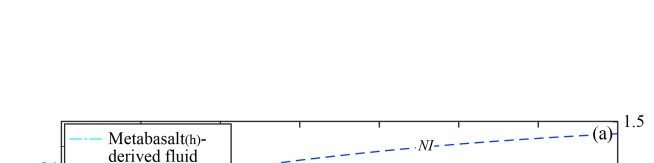

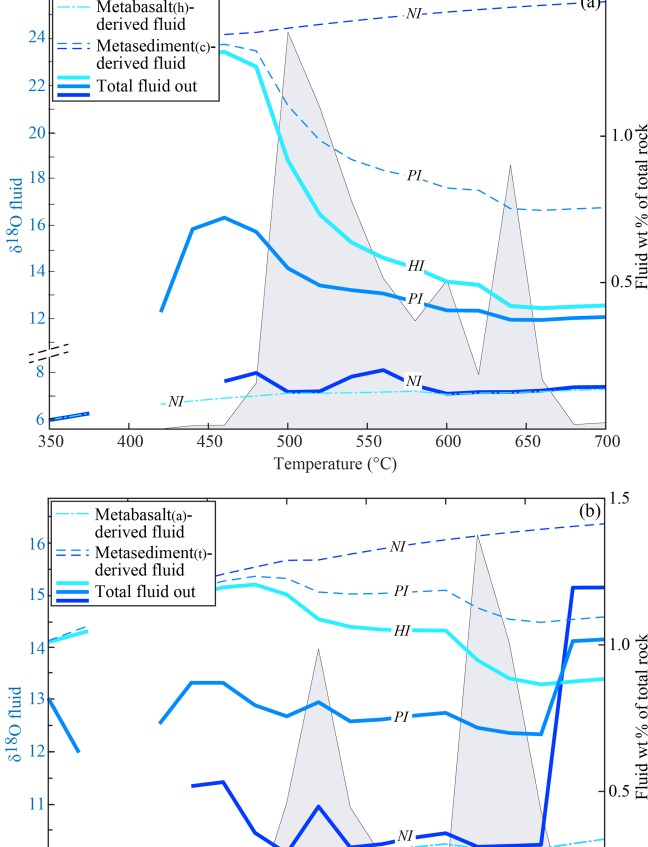

**Figure 6.** Double plot diagrams showing the oxygen isotope composition of the released fluids (left axis, coloured lines) and the amount (in wt % of the total rock) of the total fluid released by the systems (right axis, grey field) for each interaction case in the absence of serpentinite-derived fluid input. **(a)** Modelled fluid $\delta^{18}O$ values and amount for the system metabasalt$_{(h)}$ + metasediment$_{(c)}$. **(b)** Modelled fluid $\delta^{18}O$ values and amount for the system metabasalt$_{(a)}$ + metasediment$_{(t)}$. Dashed lines show the $\delta^{18}O$ values of the fluids released by each rock type and solid lines the $\delta^{18}O$ values of the final fluids released by each system. In the case of HI, all the metabasalt-derived fluid infiltrates the metasediment. Hence the final fluid released overlaps with the fluid expelled by the metasediment and only one line is represented (light blue, marked as HI). Because the $\delta^{18}O$ values of the fluids released by the metabasalts are not affected by the degree of interaction, all three cases are represented by one line (marked as NI).

higher grade where intra-crystalline diffusion becomes efficient to partially re-equilibrate garnet (Caddick et al., 2010; Lanari and Duesterhoeft, 2019). As already documented by Konrad-Schmolke et al. (2008), garnet fractionation controls the extent of the garnet stability field. Garnet crystallization is not systematically expected to occur near the peak conditions, if the matrix was strongly depleted due to garnet fractionation and the volume of garnet remains constant (i.e. for the metabasalt$_{(a)}$, Fig. 3b). While garnet fractionation is recognized to significantly affect isopleth thermobarometry and volume fractions (Lanari and Engi, 2017), its effect on oxygen isotope bulk composition and partitioning is negligible ($< 0.5 \permil$) in all the studied lithologies. In the model, the garnet fraction varies from $\sim 5$ vol. % in the metasediment$_{(c)}$ to $\sim 35$ vol % in the metabasalt$_{(h)}$ (Fig. 3), and its $\delta^{18}O$ is 0.8 ‰ to 1.7 ‰ lower than the bulk (Fig. 4a, b, e, f).

Beside garnet fractionation, dehydration due to hydrous mineral breakdown and expulsion of excess water may contribute to changing the starting chemical and isotopic bulk compositions. Baumgartner and Valley (2001) postulated that the liberation of metamorphic fluids might have a profound effect on the stable isotope composition of the residual rock. In the present study, the maximum fluid loss is from the metabasalts that release $\sim 15$ vol. % ($\sim 5$ wt %) of $H_2O$ with $\delta^{18}O$ values 0.9 ‰ to 1.5 ‰ higher than the bulk rock (Fig. 4a, b) at $T \geq 500\,°C$. This significant fluid flux produces a decrease in the bulk $\delta^{18}O$ of less than 0.2 ‰ (Fig. 4c, d). Even if more extensive dehydration occurs, the effect on the bulk $\delta^{18}O$ value will be typically lower than 1.0 ‰. No significant differences in the effect of stable assemblage evolution and phase fractionation are observed between the four lithologies. Therefore, the bulk $\delta^{18}O$ of a rock that experienced a succession of dehydration reactions, without rehydration by external fluids or major compositional changes through decarbonation or mineral dissolution, is likely to be representative of its protolith composition. In this regard, integrated thermodynamics and oxygen isotope modelling represent a key tool for quantifying the potential effects of different processes and for assessing closed- or open-system behaviours.

## 4.2 Mineral $\delta^{18}O$ zoning as indicator of open-system behaviour

In the last decades, the significant advances of oxygen stable isotope analyses by SIMS (secondary ion mass spectrometer) have allowed zoned metamorphic minerals to be analysed in situ with a precision down to 0.2 ‰–0.3 ‰ ($2\sigma$) (e.g. Ferry et al., 2014; Martin et al., 2014; Page et al., 2010). The magnitude of the intra-crystalline $\delta^{18}O$ variation in key metamorphic minerals has been widely used to establish whether a metasomatic stage is related to an internal fluid deriving from the breakdown of hydrous phases or if it reflects equilibration with an external fluid of different isotopic composition (e.g. Putlitz et al., 2000; Errico et al., 2013; Page et

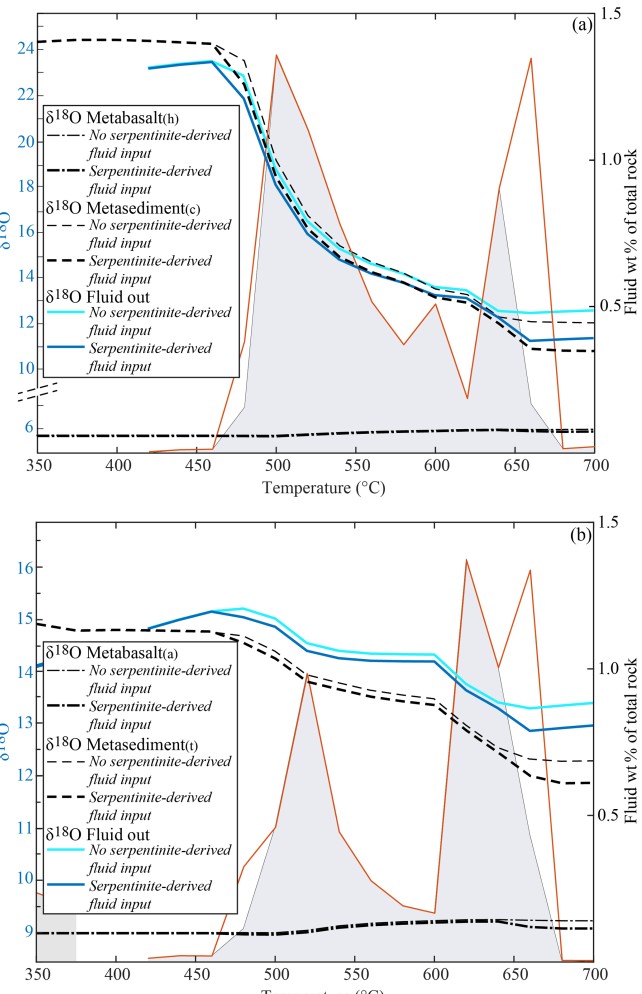

**Figure 7.** Double plot diagrams showing the effect of the fluid deriving from a layer of 150 m of pure serpentine (see text for details) on the $\delta^{18}O$ of the rock types and of the of the total released $H_2O$ (left axis) and on the amount and distribution of the $H_2O$ released by the systems (right axis). All the values are calculated assuming HI between the metabasalts and the metasediments. Black dashed lines represent the bulk $\delta^{18}O$ of the different lithologies and the blue lines the $\delta^{18}O$ of the final fluid released by the systems. The final $H_2O$ released by each system is represented with a red line. The amount and distribution of the final fluids in the case of no serpentinite-derived fluid input (grey fields) are shown for comparison. **(a)** Modelled $\delta^{18}O$ values released a CE7 fluid amount for the system metabasalt$_{(h)}$ + metasediment$_{(c)}$. **(b)** Modelled $\delta^{18}O$ values released a CE8 fluid amount for the system metabasalt$_{(a)}$ + metasediment$_{(t)}$.

al., 2013; Russell et al., 2013; Martin et al., 2014; Rubatto and Angiboust, 2015; Engi et al., 2018). Understanding the scale of fluid migration at depth and the magnitude of the interaction between fluids and minerals is of special interest and can be enhanced by modelling of such fluid flow and isotopic exchange (Baumgartner and Valley, 2001). The def-

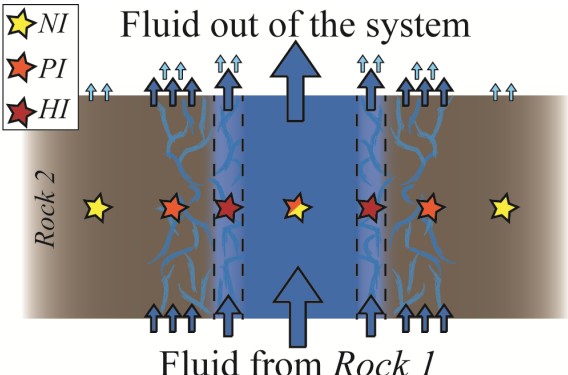

**Figure 8.** Schematic section of a channelled fluid flow where different degrees of exchange between the fluid and the rock may occur in spatial proximity. From the host rock perspective, the NI case describes the distal portion of the rock walls where no fluid infiltrates, the PI case the intermediate portion where a limited amount of external fluid is available and the HI case the pervasively infiltrated rock proximal to the vein. From the fluid perspective, the NI case describes the fluid flow in the centre of the channel, for which the exchange with the rock walls is negligible; the PI case the situation in which part of the fluid does not react with the wall rock and part equilibrates with it, and the HI case the situation in which no fluid flow occurs without equilibration with the host rock.

inition of different interaction cases (NI, PI, HI) is useful to represent various degrees of isotopic exchange between the fluid and the rock. If the flow is channelled, all the interaction cases can possibly occur in close proximity and the modelled scenarios can be linked to the evolution of different domains around the vein (Fig. 8). The flow of a pervasive fluid leads to the homogenization of the chemical potential of all components, including stable isotopes (Baumgartner and Valley, 2001), and it is represented by the HI case, as long as integrated $F/R$ ratios are high. In contrast, the flow of a channelled fluid results in local chemical heterogeneities, allowing some portions of the rock and of the fluid to remain unaffected (NI case) and some others to be only partially affected (PI case).

The first step for a meaningful interpretation of an observed intra-grain variation in $\delta^{18}O$ value is the quantification of the possible effects of changes in $T$ and mineral assemblage. Such effects are characteristic of each phase (Fig. 4a, b, e, f). Quartz, calcite and rutile are the minerals most sensitive to temperature changes. Their composition is expected to vary up to $1\%o$ per $100\,°C$, and they are stable over a wide range of temperature. For such phases, care is required in interpreting significant intra-grain $\delta^{18}O$ variations (i.e. up to $3.0\%o$) since it does not necessarily reflect interaction with an external fluid having a different isotopic composition. However, quartz and calcite are not necessarily robust minerals that can preserve chemical and isotopic zoning during metamorphism. On the other hand, the variation in $\delta^{18}O$ value over $150\,°C$ in a mineral such as gar-

net that commonly retains growth zoning is typically within $0.5\%o$ when no phase fractionation is involved and still less than $1.0\%o$ when considering the effect of previous garnet and/or excess fluid fractionation (Fig. 4a, b, e, f). Thus, any larger variation has to be linked to a significant change in bulk $\delta^{18}O$. Similar behaviour is observed for other key metamorphic minerals such as phengite, amphibole and clinopyroxene. These minerals have been widely used in metamorphic petrology as thermometers and geobarometers (Dubacq et al., 2010; Ferry and Spear, 1978; Parra et al., 2002) and are expected to be robust targets to link the fluid evolution along the $P$–$T$ path, especially when mineral relics are preserved. Due to its large capacity to preserve growth chemistry, garnet has been a primary target for microscale measurement of oxygen isotopes. Protocols and reference materials for SIMS measurements for a range of garnet compositions are well established (e.g. Martin et al., 2014; Page et al., 2010; Vielzeuf et al., 2005b) and its retentivity to high $T$ resetting by diffusion has been investigated (Higashino et al., 2018; Vielzeuf et al., 2005a). In HP rocks, various degrees of intra-grain variations in garnet $\delta^{18}O$ associated with external fluid infiltration have been reported in the literature. Where field constraints on the fluid source (i.e. oxygen isotope measurements on the feasible source lithologies) are available, the intra-grain variation can assist in the calculation of $F/R$ ratios. Martin et al. (2014) describe a shift of $-2.5\%o$ associated with infiltration of serpentinite- or altered gabbro-derived fluids in metasediments from the continental basement in Corsica. Rubatto and Angiboust (2015) observed a shift of $3.5\%o$–$4.0\%o$ in garnet from an eclogite from Monviso that they attributed to sediment-derived fluid infiltration. Vielzeuf et al. (2005b) measured a decrease of $2.5\%o$–$3.0\%o$ between garnet core and rim in the Dora Maira gneiss probably caused by the interaction with a fluid derived from the whiteschists (Gauthiez-Putallaz et al., 2016). Studies that used in situ measurement of micas are limited (Bulle et al., 2019; Siron et al., 2017), and thus the potential of mica to trace fluid–rock interaction is still underexplored. The matrix complexity of pyroxenes and amphiboles remains a challenge for SIMS measurements.

### 4.3 Interaction with fluids from an ultramafic source

The effect of pervasive fluid flow deriving from the breakdown of serpentine ($\delta^{18}O = 4.5\%o$) in a serpentine layer of different thicknesses (150, 300, 600 m, see above) on the bulk $\delta^{18}O$ of the two metabasalts is negligible ($<0.5\%o$, Fig. 7, Supplement S3). This is mainly due to (1) the minor difference in $\delta^{18}O$ between the serpentinite-derived fluid and the metabasalts ($1.2\%o$ for the metabasalt$_{(h)}$ and $4.5\%o$ for the metabasalt$_{(a)}$) and (2) the very low integrated $F/R$ ratio (0.01, 0.02 and 0.04 for the three cases). In this case, the integrated $F/R$ ratio seems to play a bigger role, since the limited change in $\delta^{18}O$ is similar for both the metabasalts even if the initial $\Delta^{18}O$ between fluid and rock is larger for

the metabasalt$_{(a)}$. With the same total volume of fluid and rock, a channelled fluid flow would imply larger volumes of fluid interacting with smaller volumes of rock (higher $F/R$ ratios) and would thus be expected to drive larger variations in isotopic composition. For instance, by increasing the $F/R$ ratio by a factor of 10 (from 0.01 to 0.1) the bulk $\delta^{18}O$ decreases by 0.6 ‰ (metabasalt$_{(h)}$) and 1.1 ‰ (metabasalt$_{(a)}$) upon infiltration of the serpentine-derived fluids.

In contrast to the metabasaltic layers, the overlying metasediments have (1) larger compositional differences with the serpentinite-derived fluid and (2) a smaller mass. The effect of the serpentinite-derived fluid input at the base of the column on the metasediment bulk $\delta^{18}O$ compositions can be up to 10 times larger than the effect that the same amount of fluid has on basaltic compositions. This is the case even when the serpentinite-derived fluid completely equilibrates with the metabasalt before interacting with the overlying metasediment. In our model, the fluid infiltrating the metasediment is always a mixture of metabasalt-derived and serpentinite-derived fluid. Considering instead an input of serpentinite-derived fluid directly in the metasediment and applying an $F/R$ ratio of 0.1, the final bulk $\delta^{18}O$ of the metasediment$_{(c)}$ decreases by 4.6 ‰ and that of the metasediment$_{(t)}$ by 2.6 ‰. These values are significant and are comparable with variations observed in natural samples. For instance, Martin et al. (2014) describe a shift in $\delta^{18}O$ of $-2.5$ ‰ among different generations of HP garnet in a sample from the Corsica continental basement (garnet mantle $\delta^{18}O = 7.2 \pm 0.4$ ‰, garnet rim $\delta^{18}O = 4.7 \pm 0.5$ ‰). The authors associate this shift with an infiltration of serpentinite-derived fluids and, to a lesser extent, altered gabbro-derived fluid. Williams (2019) describe an extreme $\delta^{18}O$ shift of $-15$ ‰ between garnet core and rim in a metasediment from the Lago di Cignana Unit. Such an oxygen isotope composition variation has been related to a strongly channelled fluid influx originating from the dehydration of serpentinites. These results demonstrate that metasediments can be a good target to detect fluids from an ultramafic source migrating upward through the subducting slab or along the subduction interface, even though the two lithologies may not be in direct contact in the field.

It is important to note that the relatively water-poor system composed of the metabasalt$_{(h)}$ and the metasediment$_{(c)}$ is more sensitive to external fluid infiltration and thus affected by the highest changes in $\delta^{18}O$, according to observations in natural systems (e.g. Page et al., 2019).

## 4.4 Effect of the subduction geotherm

As discussed in detail by previous studies (Baxter and Caddick, 2013; Hacker, 2008; Hernández-Uribe and Palin, 2019; Syracuse et al., 2010), the subduction geotherm has an important effect on hydrous phase stability and $P–T$ conditions of fluid release into the mantle wedge. Along the average D80 geotherm by Syracuse et al. (2010) (Fig. 2), the top of

the slab crust releases $\sim 95$ % of the water at $\sim 80$ km, during the transition from partial to full coupling. Along the cold geotherm by Penniston-Dorland et al. (2015), the first significant fluid pulse (20 %–40 % of water released) occurs as a consequence of the breakdown of glaucophane and actinolite at greater depths than predicted by our model ($\sim 500$ °C and 2.7–2.8 GPa, 90–100 km depth), and the remaining water is released at depth $> 100$ km. Both those models imply a relatively dry mantle forearc region, contradicting what has been described by Bostock et al. (2002). By contrast, along the warm geotherm by Penniston-Dorland et al. (2015), the breakdown of chlorite, epidote and actinolite releases 40 %–50 % of the water at 460–470 °C and $\sim 0.6$ GPa ($\sim 20$ km depth); after this stage, $\sim 95$ % of the water is released at depth $< 70$ km. This implies that little to no water is available at subarc depths. The average geotherm by Penniston-Dorland et al. (2015) is hotter than the one by Gerya et al. (2002) used in our model for $T < 650$ °C and $P < 2.5$ GPa (Fig. 2). Along this $P–T$ gradient, chlorite is the main water carrier in low-grade conditions and the first two main dehydration pulses occur at depths of 30–40 km (significant decrease in chlorite, 20 %–25 % of water released) and of $\sim 50$ km (complete breakdown of chlorite, 20 %–25 % of water released). Along this geotherm, the most water is released at shallower depths than in our model. Differences can be investigated in detail by modelling each case with PTLoop. Nevertheless, the effects of fluid–rock interaction on the bulk and mineral $\delta^{18}O$ compositions follow the general trends described above. Different parageneses are expected to form during hydration, but the shift in bulk $\delta^{18}O$ remains constrained by the $F/R$ ratio and the isotopic composition of the incoming fluid.

## 4.5 Implication for mantle wedge hydration

Infiltration of the slab-derived fluid into the mantle wedge is important for subduction zone settings because mantle minerals are strongly depleted in volatiles. At equilibrium, a free aqueous fluid is not stable in the mantle wedge at $T < 650$ °C until a fully hydrated mineral assemblage has formed (i.e. serpentine, chlorite, talc, and amphibole) (Manning, 2004). As shown above, the $P–T$ conditions of $H_2O$ release from the subducting slab, as well as the volume and the $\delta^{18}O$ of the liberated $H_2O$, can vary according to the geometry of the subduction zone and the composition of the subducting lithosphere (e.g. Hacker, 2008; Hernández-Uribe and Palin, 2019; Poli and Schmidt, 2002). The program PTLoop allows the $P–T$ conditions, amount and oxygen isotope composition of the released fluid to be calculated for the system of interest. The presented model uses an intermediate $P–T$ gradient that stabilizes lawsonite and results in a first significant fluid pulse at 65–70 km depth and a second pulse at $\sim 80$ km depth. Below that depth, phengite is the main carrier of $H_2O$. This implies that the majority of fluid is released in the forearc region, in agreement with previous studies investigating

the dehydration of basaltic and sedimentary components of the slab (e.g. Baxter and Caddick, 2013; Kerrick and Connolly, 2001; Schmidt and Poli, 1994).

The influence of the slab-derived fluid on (1) the degree of hydration and (2) the $\delta^{18}O$ modification of the overlying mantle rocks was estimated on the basis of the results presented above. A slab composed by the metabasalt$_{(a)}$ and the metasediment$_{(t)}$ (left column in Fig. 1a, assuming the real thickness of the slab to be 3 times the modelled one) subducting at $1\,\mathrm{cm\,yr^{-1}}$ was considered (Fig. 9a). This subduction rate represents a conservative estimate, considering that previous averages have proposed $4$–$5\,\mathrm{cm\,yr^{-1}}$ (Stern, 2002). Any faster subducting rate may shorten the timescale of the processes discussed below, allowing larger fluid fluxes in the mantle wedge over the same interval. Mechanical decoupling between the slab and the wedge and a steady-state cold mantle wedge are assumed (e.g. Abers et al., 2006; Hirauchi and Katayama, 2013; Wada et al., 2008). The fluid released by the slab at $500$–$520\,^\circ\mathrm{C}$ with a characteristic $\delta^{18}O = 15.0\,‰$ (Fig. 6a) was allowed to infiltrate into an initially dry peridotite (composition KLB-1 from Walter, 1998, simplified to the Fe–Mg–Al–Si system; Table S2.4 in Supplement S2) with $T = 570\,^\circ\mathrm{C}$ and an initial $\delta^{18}O = 5.5\,‰$ (Eiler et al., 1997; Mattey et al., 1994). This simplified model ignores the dynamics of fluid infiltration and assumes pervasive flow. Given the column length of 1 m, a subduction rate of $1\,\mathrm{cm\,yr^{-1}}$ implies that, in 100 years, any fixed point (i.e. fixed $P$–$T$ conditions) at the slab–mantle interface receives the total amount of fluid that a single column can liberate in those conditions. Hence, in this example 4892.6 kg of water per 100 years (i.e. the amount released by the considered column at the chosen conditions, see above) infiltrate the mantle wedge. The released fluid first interacts with a small volume at the slab–mantle interface (V1 $1000 \times 1000 \times 1$ m; Fig. 9b) and once it has equilibrated and saturated V1, it infiltrates volume V2 ($3000 \times 2000 \times 1$ m; Fig. 9b). Both the volumes were scaled to 1 : 3 for the calculation in order to maintain the volume proportion with the modelled slab. The slab-derived fluid, which is in continuous supply during subduction of new material, infiltrates V1 and causes a progressive change in mineralogy from olivine + orthopyroxene + garnet to serpentine + chlorite + minor olivine until the rock reaches saturation after 0.35 Myr of subduction. At this stage, V1 has a bulk $\delta^{18}O$ of $\sim 8\,‰$ that is significantly higher with respect to the initial mantle value (Fig. 9c). With ongoing subduction, the continuing release of water from new slab material under a static mantle drives the $\delta^{18}O$ of the volume of mantle wedge toward higher values. The water that will infiltrate V2 has a $\delta^{18}O$ that depends on (1) the $\delta^{18}O$ of the slab-derived water and (2) the buffering capacity of V1. The same changes in mineral assemblage described for V1 occur also in V2, while the bulk $\delta^{18}O$ of V2 increases more moderately than the one of V1 and reaches a $\delta^{18}O$ of $\sim 6\,‰$ after 0.75 Myr of subduction (Supplement S4).

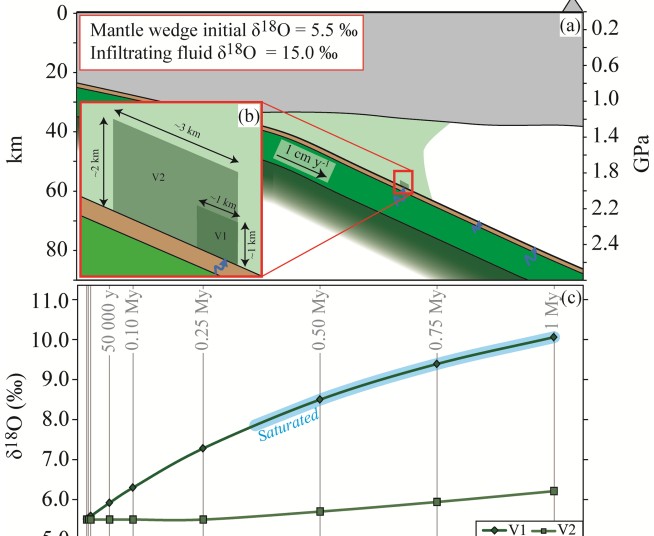

**Figure 9.** Case model for mantle wedge hydration. **(a)** Sketch of a subduction zone (crustal thickness and serpentinized mantle wedge dimensions are from Bostock et al., 2002 CE9. The subducting slab is composed of metabasalt$_{(a)}$ and metasediment$_{(t)}$ as shown in Fig. 1a, left column. **(b)** Geometry of the model. The blue arrow represents slab dehydration at $500$–$520\,^\circ\mathrm{C}$. Abbreviations: V1 represents the volume of mantle rocks at the interface and V2 the surrounding volume (see text for details). **(c)** Plot of the bulk $\delta^{18}O$ variations of V1 and V2 as a consequence of continuous slab dehydration over 1 Myr.

In the proposed model, most of the fluid is released by the slab at forearc depths. However, in most subduction zones no melting appears to occur in the forearc region and the serpentinite acts as the effective $H_2O$ absorber (Iwamori, 1998), recording the possible variation in $\delta^{18}O$ induced by the slab-derived fluid. Progressive oceanward migration of the slab ("slab rollback") has been regarded as an important mechanism acting in most active subduction zones (e.g. Heuret and Lallemand, 2005; Nakakuki and Mura, 2013). The rollback of the slab results in a lateral extension of the serpentinized wedge. As a consequence, the melt ascending below the arc can interact with serpentinized, high-$\delta^{18}O$ mantle portions that were originally part of the forearc mantle and modify its original isotopic composition. High-$\delta^{18}O$ arc lavas and melt in arc-setting peridotites have been described (e.g. phenocrysts in lavas from central Kamchatka: olivine $\delta^{18}O = 5.8\,‰$–$7.1\,‰$ and clinopyroxenes $\delta^{18}O = 6.2\,‰$–$7.5\,‰$, Dorendorf et al., 2000; New Guinea: silicate glass inclusions in olivine $\delta^{18}O = 8.8\,‰$–$12.2\,‰$, clinopyroxenes in metasomatized lehrzolite $\delta^{18}O = 6.2\,‰$–$10.3\,‰$, Eiler et al., 1998), but the mechanism of crustal contamination is still debated. Our results support the model proposed by Auer et al. (2009) that relates such high-$\delta^{18}O$ lavas to the interaction between primitive basaltic melts with an

uppermost mantle that was hydrated and enriched as part of the forearc mantle prior to trench migration.

## 4.6 Model applications and future directions

The presented approach has a broad range of applications for modelling fluid–rock interaction in different tectonic settings. We have presented here an example of subducted crust but the same principles apply also for regional metamorphism or hydrothermal systems. The model also provides new ways to quantify the degree of interaction of an external fluid within the same rock unit. We have shown that the observed effect on the $\delta^{18}O$ of a rock of channelled vs. pervasive hydration is strictly coupled with the composition of the fluid source. Nevertheless, important insights can be given by linking observations of ideal cases with modelling even if the composition of the infiltrating fluid is not known a priori. For instance, the oxygen isotope composition of a fluid source can be retrieved when a variation in $\delta^{18}O$ is observed within the same rock type from the more hydrated to the less hydrated portions, even in the absence of a clear presence of a vein or vein system. Alternatively, the degree of equilibration of the host rock around a vein can be calculated when the isotopic composition of the fluid is known. Therefore, the combination of mineral-scale in situ oxygen isotope analyses with major and trace element mapping will provide much more detailed information on quantitative element mobility during fluid–rock interaction.

Fluids play a central role as a catalyst for chemical reactions in rocks. Generally re-equilibration reactions occur only in the presence of fluids that either derive from breakdown of hydrous phases or from external sources (e.g. Airaghi et al., 2017; Cartwright and Barnicoat, 2003; Engi et al., 2018; Konrad-Schmolke et al., 2011; Rubatto and Angiboust, 2015). The program PTLoop calculates at which $P$–$T$ conditions the breakdown of hydrous phases occurs, and, consequently, metamorphic reactions and free fluid are expected. If fluid-driven reactions occurred in a rock at $P$–$T$ conditions where no release of internal fluid is expected, the role of an external fluid should be considered, and its amount and isotopic composition can be retrieved using the approach outlined in this paper.

Fluid–rock reactions in subducted lithosphere are likely to involve more complex open-system processes than the dehydration and rehydration considered for this model, driving silicate and carbonate dissolution, transport and reprecipitation (e.g. Ague and Nicolescu, 2013; Piccoli et al., 2016). Carbon release via the dissolution of calcium carbonate has been recognized to have important implications for $CO_2$ release from subduction zones, and it is controlled by $H_2O$-rich fluid infiltration (e.g. Ague and Nicolescu, 2013; Frezzotti et al., 2011; Gorman et al., 2006; Kerrick and Connolly, 2001). Future models may account for such variations in reactive major element bulk composition of the rocks along the $P$–$T$ evolution as a consequence of mineral net

transfer reactions occurring simultaneously with fluid–rock exchange (Baumgartner and Valley, 2001), in addition to water liberation and mineral fractionation (see above). This would require additional considerations on the changes in the fluid isotopic composition during transfer of solute species through the fluid. The implementation of other fluid species beside $H_2O$, such as $CO_2$, could be assessed, provided that (1) reliable constraints on the oxygen isotope fractionation between these species and water or minerals are determined and (2) their consistency with other available data is established. Other species such as $CH_4$ and $H_2$ do not contain any oxygen, and thus they are likely to be less relevant to this model.

## 5 Conclusions

We developed a user-friendly tool that combines equilibrium thermodynamic with oxygen isotope fractionation modelling for investigating the interaction between fluids and minerals in rocks during their metamorphic evolution. The program simulates along any given $P$–$T$ path the stable mineral assemblages, bulk $\delta^{18}O$ and $\delta^{18}O$ of stable phases and the amount and oxygen isotope composition of the fluid released.

The capabilities of the program PTLoop are illustrated by an application to subduction zones, but the presented modelling strategy can be applied to various metamorphic and tectonic settings. In this study, the chosen system represents a section of subducting oceanic crust composed by a lower layer of metabasalt and an upper layer of metasediments of carbonaceous or pelitic composition. The calculation follows a step-wise procedure along the chosen $P$–$T$ path. During the prograde evolution, any mineral and excess fluid can be fractionated from the reactive bulk composition.

Mineral fractionation and/or excess fluid loss produce only minor (i.e. $< 1.0\,‰$) shifts in the bulk $\delta^{18}O$ of any lithology. Hence, the bulk $\delta^{18}O$ of a rock that experienced a succession of such processes without interaction with external fluids is likely to be representative of its protolith composition. Variations in $\delta^{18}O$ of stable phases due to mineral fractionation and/or excess fluid loss are also negligible (i.e. $< 0.5\,‰$), while the effect of temperature variation over a range of $\sim 150\,°C$ on the mineral $\delta^{18}O$ is phase dependent and may be significant ($> 1.0\,‰$).

Interaction with an external fluid of different oxygen isotope composition leads to shifts in bulk and mineral $\delta^{18}O$ values according to the degree of fluid–rock interaction and $\delta^{18}O$ difference between the rock and the fluid. Extremely large variations in bulk $\delta^{18}O$ of $\sim 12\,‰$ are calculated for the carbonate metasediment equilibrating with a fluid derived from a metabasalt with an initial hydrated MORB composition, while small variations of $\sim 3\,‰$ are calculated for the terrigenous metasediment equilibrating with a fluid from a metabasalt that derives from an altered oceanic crust. When

50 % or more of the fluid deriving from dehydration of the metabasalts equilibrates with any of the overlying metasediments, the final $\delta^{18}O$ of the fluid released by the system has a dominant sedimentary signature, with values between 12‰ and 18‰. Such fluids have $\delta^{18}O$ values significantly higher than the mantle value (5.5‰) and have a great potential to modify the oxygen isotope composition of the mantle wedge at the slab–mantle interface. Extensive serpentinization and a $\delta^{18}O$ increase of ∼ 2.5‰ are modelled at the interface already after 0.35 Myr of ongoing subduction.

PTLoop provides a powerful way to evaluate the effect of closed-system vs. open-system behaviour with respect to oxygen isotopes during the evolution of the rocks. Different degrees of interaction between the external fluids and the sink lithology can be simulated and the effects of internally vs. externally buffered fluids on the mineral paragenesis and on the mineral isotopic composition investigated.

Measured oxygen isotope compositions in minerals, intra-grain or bulk $\delta^{18}O$ variations at different scales can be compared with the results of the model for specific scenarios. If the measured isotopic compositions are not consistent with the behaviour of a closed system, the presented approach can be used to determine feasible external fluid sources, to estimate the degree of fluid–rock interaction and the metamorphic conditions at which this happened. This modelling strategy can also assist in retrieving the oxygen isotope composition of a fluid source when a variation in $\delta^{18}O$ is observed within the same rock type from the more hydrated to the less hydrated portions, even in the absence of a clear presence of a vein or vein system. Our model thus opens new avenues for mapping fluid pathways related to external fluid infiltration during the metamorphic evolution of the crust, with important consequences for element recycling in subduction zones and the investigation of fluid-induced earthquakes.

*Code availability.* A compiled version of the program PTLoop, the oxygen isotope fractionation database and the thermodynamic database used for this study are available at http://oxygen.petrochronology.org (Lanari and Vho, 2020).

*Supplement.* The supplement related to this article is available online at: https://doi.org/10.5194/se-11-1-2020-supplement.

*Author contributions.* AV developed the model and performed the calculations. PL supervised the software development and contributed to the code implementation. DR and JH contributed to formulate and design the model and to the interpretation of the results. AV prepared the paper with contributions from all co-authors. DR conceived the project and secured funding.

*Competing interests.* The authors declare that they have no conflict of interest.

*Special issue statement.* This article is part of the special issue "Exploring new frontiers in fluids processes in subduction zones". It is a result of the EGU Galileo conference "Exploring new frontiers in fluids processes in subduction zones", Leibnitz, Austria, 24–29 June 2018.

*Acknowledgements.* We thank the conveners and the participants of the EGU Galileo conference "Exploring new frontiers in fluid processes in subduction zones" for constructive discussions. We also thank Ulrich Linden for the technical support and the reviewers Ralf Halama and Alberto Vitale Bovarone for the constructive comments.

*Financial support.* This research has been supported by the Swiss National Science Foundation to Daniela Rubatto (grant no. 200021_166280) and Jörg Hermann (grant no. 200021_169062).

*Review statement.* This paper was edited by Nadia Malaspina and reviewed by Ralf Halama and Alberto Vitale Brovarone.

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

**Remarks from the language copy-editor**

CE1     Please note that this is our house standard.

CE2     If this is the prefix referring to a small size, "nano" is the correct spelling. If the meaning is different here, please let us know, as this would be useful for us to note for future reference. Thanks!

CE3     Written out for clarity.

CE4     Need approval form editor.

CE5     This change need the approval from the editor (although it is already inserted).

CE6     Change in section title needs to go to editor.

CE7     Need approval from editor.

CE8     Need approval from editor.

CE9     Thanks for clarifying the copyright situation. We have now adjusted capitalisation to our house standards in the figure.

**Remarks from the typesetter**

TS1     Adjusted to our house standards (not italic).

TS2     Please check equation.

TS3     Adjusted to our house standards. Abbreviations should be roman.

TS4     Adjusted to our house standards for percent and permil.

TS5     Please provide initial.