# Peer review of "Tracing fluid transfers in subduction zones: an integrated thermodynamic and $\delta^{18}$ O fractionation modelling approach"

_Solid Earth, 2019_

## Referee Comment (RC1) · Ralf Halama (Referee) · 8 Oct 2019

Manuscript se-2019-140
Journal: Solid Earth
Special Issue: Exploring new frontiers in fluid processes in subduction zones
Authors: Alice Vho et al.
Title: Tracing fluid transfers in subduction zones: an integrated thermodynamic and $\delta^{18}O$ fractionation modelling approach.
Review by Ralf Halama (Keele University), 8/10/19

**General comments:**

This manuscript presents a new approach of combined thermodynamic and oxygen isotope fractionation modelling applied to fluid-rock interaction in subduction zones. It is an important step forward in using oxygen isotopes as a tool to track fluid-rock interaction because the oxygen isotope compositions of minerals and fluids can be predicted taking into account successive changes in pressure-temperature conditions. This approach can also be applied to other geological settings and the results of such modelling are relevant for interpreting oxygen isotope data of natural samples.

I have made several specific comments that should be useful to improve clarity and consistency in the text. One aspect that can be expanded upon, in my opinion, are the implications for interpreting oxygen isotope data of natural samples and providing links to observed oxygen isotope variations and the interpretations made thereupon.

At the date of review, the database, which is used for the modelling and which is fundamental for the models, is currently under review for another journal and can hence not be accessed. This is somewhat problematic regarding the final publication of this manuscript, at which point the database should be accessible to readers (see comment below).

**Specific comments:**

10: For clarity, it should be added the Gibbs energy minimization is carried out for given whole-rock compositions.

15-20: The abstract would become clearer if a structure similar to the order/sections in the manuscript is adopted. For instance, the infiltration of an external fluid into mafic rocks is mentioned as point (1) in the abstract, but it is the last section in the results (3.3.4) and also near the end in the discussion. Better to first describe the simple case of dehydration only ("no interaction" case); it should also be mentioned that dehydration in the metasediments is also considered in the models. Second, the influx of MORB-derived fluid should be addressed; currently, MORB-derived fluids are not explicitly mentioned in the abstract, so this should be added as this process features prominently in the results and discussion. Finally, the influx of fluid derived from ultramafic rocks can be mentioned.

82-90: Since sediment dehydration is also considered, this should be explicitly mentioned here. It is nicely described in the figure caption to Fig. 1 that the fluid

produced in the "No Interaction" case is a mixture of MORB-derived and sediment-derived fluid.

148: Reference to the database in Vho et al. (in review). As a reviewer of this manuscript for Solid Earth, I do not have access to this database and do not know the current status regarding publication of this database. This is somewhat problematic as the database is fundamental for the modelling performed in this manuscript. I am convinced that the authors are pursuing the publication of this accompanying manuscript as quickly and efficiently as possible, but I would like to see this the accompanying manuscript as accepted manuscript with doi number before the final version of this manuscript for Solid Earth goes online. The reason is that readers should be able to adequately follow and reconstruct all the information and proceedings of this article. This is difficult if the database is not accessible and may become problematic if the accompanying manuscript is never published (I do not expect this to happen, but it seems a sensible approach notwithstanding the impeccable reputation of the present author team).

194: As these fractionation factors are explicitly mentioned, it would be useful to specify for which temperature the fractionation of 2 ‰ is representative of.

203: white mica: The phase diagrams in Fig. 3 show "ph" (phengite), but in the text mostly "white mica" is used (but see line 212). Does "white mica" always stands for a potassic white mica? For consistency, only one term should be used throughout, and an explanation regarding the composition of the white mica would also be useful (e.g. the modelled composition might be a typical muscovite at lower grades but becomes more phengitic at higher temperatures and pressures).

212-224: There are a few minor issues in this section that should be addressed:
a) Modal phase changes are referred to partly in a neutral way (increase/reduction) but in other cases specific reactions are invoked (e.g. gln consumed in favour of jd+ank, lws breakdown producing grt). From a perspective of a metamorphic petrologist, one would be interested to see the full reaction equations. Two examples: If ankerite is produced, carbonate or $CO_2$ is required as reactant; which other phases are involved in the lawsonite breakdown reaction? However, in the context of this study, this detail may not be necessary, and it may be suffice to formulate in a way without referring to specific reactions.
b) Clinopyroxene composition: As the change from omphacite to jadeite is mentioned, please clarify at which compositional boundary (mol% jadeite component) the change in name is made. Or is it a pure end-member jadeite? The coexistence of jadeite and omphacite should also briefly be addressed as in natural rocks, one would presumably expect only one clinopyroxene with changes in the jadeite-component in omphacite. The authors point out that the co-existence of two amphiboles is of little relevance for the oxygen isotope modelling – is this similar for the pyroxenes? This should be clarified.

227: Initial water-saturated conditions: Please explain and justify the choice of water-saturated conditions, in particular for the fresh MORB. If one assumes that fresh MORB is initially composed of nominally anhydrous minerals only, where does the water come from? In the discussion later, the water released from the slab is dominated by the MORB-derived fluid, and presumably this is due to amounts of water stored under water-saturated conditions initially. Hence, does this initial assumption affect (at least some) of the model calculations, and how large is the effect? I appreciate that not all

possible scenarios can be addressed in a single manuscript, but a brief justification of the choices made would be useful.

234: Glaucophane and actinolite and the intermediate fluid pulse: It seems in the figure that the growth of talc takes up the water released by consumption of actinolite and glaucophane, as the modal proportion of talc increases from 580 to 600°C, whereas the amount of water appears to increase at >600 °C when the modal proportions of talc and then lawsonite decrease. Please check carefully and modify the text accordingly.

236: The liberation of water from the carbonate sediment is specified, but release of $CO_2$ is not mentioned. Does any release of $CO_2$ occur? Carbonate phases appear to remain stable, but the aspect should still be briefly explained for clarity.

282: Mafic fluid (see also 288, 297, 300 and elsewhere): Using the terms mafic fluid and ultramafic fluid is not appropriate and should be avoided. The term "mafic" is derived from magnesium and ferrum (iron) rich, which is appropriate for rock compositions but not for the fluids considered here. The same applies to "ultramafic fluids" (e.g. in lines 309, 311 and elsewhere), serpentinite-derived fluid should be used instead.

311: This statement is a bit vague. What exactly is the effect in the PI and NI cases on MORB? If the variations in the sedimentary rocks decrease to zero, does that mean there is no effect at all, or no change compared to the previous cases? Please formulate more precisely here.

331-337: The example of the granite appears to be out of place here, as granite has not been considered anywhere else in the manuscript. A dry basalt would be a more appropriate example, which can be linked to the scenarios considered much better. But the results presented show the limited effect on the O isotope variation anyway, so consider deleting this section altogether.

363-364: This statement is important, and could be highlighted in abstract and/or conclusions.

387-393: This section would benefit from a few more details regarding the studies on oxygen isotope zoning in metamorphic minerals, and how the modelling results can be linked to these results (and possibly used to support interpretations or argue for alternative interpretations). Questions that are of interest to the reader include: What kind of zonation was observed in the minerals studied? With which of the modelled scenarios do these patterns coincide? Providing more details here and some specific examples would also be useful to emphasize the wider implications for studies based on natural samples.

403: Integrated Fluid/rock ratios: It is not entirely clear where the numbers come from as they have not been mentioned before. Please clarify.

410: Serpentinite-derived fluid input into the sedimentary layer: Is this fluid in the models not a mixture of serpentinite-derived fluid and MORB-derived fluid since MORB also dehydrates? If so, clarify this point.

413-415: Detection of serpentinite-derived fluids: It would be useful if the authors could refer to the (possible) detection of such fluids in real sediments to underline the relevance of their study.

416-418: Effects: The relatively "dry" system still starts with a water-saturated MORB; so the reader may wonder how things change if the system is really dry – would the oxygen isotopic effects even larger? (see also earlier comments).

419-437: This section seems rather unnecessary because it does not add much to the discussion on oxygen isotopes, the main statement emphasizing that the trends are similar to the ones shown earlier. The discussion on water release is fine but key points could be incorporated into the section "Model geometry" where some of the differences between the P-T paths are already highlighted.

450-468: Can the relevant equations that consider the subduction rate and the volumes of fluid released be shown here so that readers get a better understanding of the modelling approach?

475: What exactly are "high" $\delta^{18}O$ arc lavas. Please provide some values or a range of values.

486: Another important statement relevant for the interpretation of natural samples, which may also be emphasized in the conclusions.

515: Interesting aspect which may be of interest to studies on natural serpentinites. For instance, have such elevated O isotope signatures been observed in natural wedge serpentinites? Or can O isotopes be used to distinguish wedge from abyssal peridotites in the geological record? Briefly expanding on these aspects would widen the relevance of this study.

528-530: I have not checked or tested this version since I am not a user of Matlab.

Figures:
Figure 1: As in the text, please avoid the terms "mafic fluid" and "ultramafic fluid". Add $\delta^{18}O$ to the numbers given in the figure. Clarify that 4.5 ‰ is a fluid value, not the value of the serpentinite. Give the sources for the $\delta^{18}O$ values used in the figure, or refer to the text.
Figure 2: The meaning of the abbreviation D80 should be explained. Moreover, one may wonder whether an average geotherm is useful as it may not represent any real subduction zone. Typo in the figure "Syracuse". Regarding the lines in this figure and in other figures, they are dashed rather than dotted and should be labelled accordingly.
Figure 3: Mineral abbreviations should be explained. It would also help to indicate initial water contents in the figure or the caption.
Figure 4: The line for titanite is almost invisible in a print out, a somewhat darker colour would improve visibility. For diagrams (g) and (h), I recommend presenting separate diagrams for the partial and high interaction cases because the distinction of the lines marked with stars is not very clear (the bulk trend could be copied into the respective other diagram for comparison). As above, lines are dashed (short bars) rather than dotted (points).
Figure 6: Avoid terms "mafic" and "ultramafic" fluid.

**Technical corrections:**

13: composed of

14: assemblages (plural, since mafic crust and sedimentary cover are considered)

28: reactions (plural)

45: An alterantive approach follows what has been …

48: Such an approach

52: fluid/rock ratios (plural)

56: in the last two

67: Delete "on average"; instead: "lithosphere is typically composed of a section of igneous oceanic crust …" (adding igneous helps to clarify the sedimentary cover is considered separately).

78: Replace "is due to" with "was chosen to take into account" (to clarify that it was intentionally chosen).

79: unsatisfactory models (adjective)

83: migrates

87, 88: crust-derived (hyphen required because words cannot stand on their own)

101: P and T are already used in line 58, so the abbreviations should be explained there (that is, where they are first used in the text).

108: accounted for in the …

143: Typo: "the its" – please correct

172: Replace "following" by "subsequent"

205-207: Sentence structure: For either composition, the volume of … decreases from 480 ℃ and 1.90 GPa until complete …

208: Replace "higher conditions" by "higher grades" or "higher P-T conditions"

229: 2.60 should probably be 2.06?

235: 2.03 should probably be 2.30?

253: Typo: starting

275, 276: Do these changes occur over the total temperature range modelled? Please specify.

282: decrease by

283: decrease by a maximum of

288: most of the fluid

318: decreases by

350: extent

393: measurements of oxygen isotopes

395, 396: Check citations Vielzeuf et al. (2005 or 2005a and 2005b).

426: what has been described

483: Replace extensive with pervasive.

503: "carbonaceous" instead of carbonatic

785, 787: "mixture" instead of "mixing"

826: all three

---

## Short Comment (SC1) · 11 Nov 2019

The manuscript by Vho et al presents a new phase equilibria-$\delta$18O model capable to predict $\delta$18O variations in rocks and fluid, potentially at a very large scale, in subduction zones. There is no doubt that this work is timely and could have a significant impact on an international scientific audience. There is also no doubt that this work represents an important step forward towards understanding fluid-rock exchanges in subduction zones (and not only). Although, in my opinion, additional steps will be needed to improve this tool in the future, this early work certainly represents an excellent start. My sincere congratulations for the great work. Below I will list a series of issues that are

[Figure]

not considered (or not clearly explained) in the text, and that may potentially impact the results and interpretations of the modeling (I think they really do impact the results). For the sake of clarity, I am not suggesting modifying the code or the presented models for this paper, but rather to clarify that these missing (?) issues are part of the assumptions of this early study. In my opinion, this would avoid unnecessary criticisms during this fundamental stage of development.

- Chemical system: carbon is not present in the list at line 164. It certainly exists in the models (carbonate present), but it is not clear if $CO_2$ (as well as other species such as $CH_4$ and $H_2$) in the fluid was considered or forced not to form. It is clear, however, that $CO_2$ and other species (if any) were not considered in the $\delta$18O budgets (line 118). Same for S. A sentence should be added. - The text should clarify if $CO_2$ was considered as a negligible parameter in this model (non just not considered). To be honest, I do not see how percolation of potentially high fluid fluxes through the carbonate layer should not mobilize (not just equilibrate) a large portion of the bulk carbonate O. Take the example of Ague and Nicolescu (2013 Nat Geo): an almost complete carbonate devolatilization along a fluid channel. Or the reverse carbonation (Piccoli et al 2016; Scambelluri et al 2016). Can O-bearing fluid species other than $H_2O$ modify the model assumptions? If yes (e.g. Baumgartner and Rumble), something should be said. If not, why? The sentence at line 118 is not enough in my opinion and a more detailed presentation of the related biases should be provided. - There is no mention to the potential effect of evolving redox (e.g. when $H_2O+CH_4$ go to $CO_2 + H_2$) on the $H_2O$ $\delta$18O. Of course, the cap delta between $H_2O$ and minerals would not change, but the relative signatures would. This should be at least mentioned and/or justified. This is relevant because, for example, in the terrigenous layer, a fluid in equilibrium with graphite (not considered in the model) may be strongly enriched in one or the other C-bearing species relative to $H_2O$. - Still on line 118: although the choice of considering molecular fluid species only does not fully reflect the technical means we have today (e.g. DEW model), I agree that this is probably the right choice for this early contribution. However, especially because this study centers on fluid-rock interactions

and metasomatism, the effect of omitting ionic species and their effect of potentially large mineralogical/mass changes has to be introduced. The manuscript cites a series of natural examples of strong fluid-mediated O resets. These rocks are in most cases associated with dramatic major element variations that cannot be explained without species more complex than molecular H2O. The possibility that these species have a negligible effect on the $\delta$18O of the system has to be demonstrated. For example, the cap delta between HCO3- and H2O at room T is about 40‰At higher T it should be lower, but maybe still significant if present in large amounts. At least for the carbonate layer, species like HCO3- may be important at the considered conditions (see Facq et al 2014 GCA). Here again I suggest providing more details on these assumptions and potential biases. See also the potential effect of pH on stable isotope variations (Ohmoto 1972). - F/R ratios. The only values of F/R ratios that I could find in the text (apologies if I am wrong) appear very low to me, especially in the case of channelized fluid flow. As time is present in the proposed model, it could help having some idea on how the proposed fluid/rock ratios translate into time-integrated fluid fluxes. The proposed values should at least in part correspond to the time-integrated fluid fluxes estimated in pervasive vs. channelized fluid systems in crustal settings (see review by Ague 2014 for example). F/R ratios alone do not provide insights on the hydrology of the systems and are sometimes meaningless (Baumgartner and Ferry 1991). I understand that many times this choice is imposed by the numerical code itself, but here you have the means to do this conversion at least once in the text, for reference. This could be also introduced at line 52.

Line 19: bulk $\delta$18O value: in the source? Line 85: and also on the fluid speciation... that is not considered here but that can strongly modify the $\delta$18O evolution of the fluid/rock system. For example, at 500 °C, the Cc-H2O and CC-CO2 cap delta for O differ by about 6‰Line 110: "excluding" is misleading in my opinion. You mean removing from the reactive bulk, right? 161: can you clarify the meaning of natural profiles? Line 181-182: do the chosen values take into account processes like decarbonation? 193: this sounds like a model-driven assumption. Could you clarify? 323: $\delta$18O of the

water: this is still a model assumption. I would say fluid instead. 344: increase in bulk $\delta$18O: increase in the reactive bulk $\delta$18O? 345: reactive bulk $\delta$18O? 361-366: Here is where I miss the effect of decarbonation/dissolution and species other than H2O in the model. I suggest adding a sentence to recall the assumptions. 492: Airaghi et al: I suggest adding a couple more reference on this topic.

---

## Referee Comment (RC2) · Alberto Vitale Brovarone (Referee) · 22 Nov 2019

The manuscript by Vho et al presents a new phase equilibria-$\delta$18O model capable to predict $\delta$18O variations in rocks and fluid, potentially at a very large scale, in subduction zones. There is no doubt that this work is timely and could have a significant impact on an international scientific audience. There is also no doubt that this work represents an important step forward towards understanding fluid-rock exchanges in subduction zones (and not only). Although, in my opinion, additional steps will be needed to improve this tool in the future, this early work certainly represents an excellent start. My sincere congratulations for the great work. Below I will list a series of issues that are

[Figure]

not considered (or not clearly explained) in the text, and that may potentially impact the results and interpretations of the modeling (I think they really do impact the results). For the sake of clarity, I am not suggesting modifying the code or the presented models for this paper, but rather to clarify that these missing (?) issues are part of the assumptions of this early study. In my opinion, this would avoid unnecessary criticisms during this fundamental stage of development. - Chemical system: carbon is not present in the list at line 164. It certainly exists in the models (carbonate present), but it is not clear if CO2 (as well as other species such as CH4 and H2) in the fluid was considered or forced not to form. It is clear, however, that CO2 and other species (if any) were not considered in the $\delta$18O budgets (line 118). Same for S. A sentence should be added. - The text should clarify if CO2 was considered as a negligible parameter in this model (non just not considered). To be honest, I do not see how percolation of potentially high fluid fluxes through the carbonate layer should not mobilize (not just equilibrate) a large portion of the bulk carbonate O. Take the example of Ague and Nicolescu (2013 Nat Geo): an almost complete carbonate devolatilization along a fluid channel. Or the reverse carbonation (Piccoli et al 2016; Scambelluri et al 2016). Can O-bearing fluid species other than H2O modify the model assumptions? If yes (e.g. Baumgartner and Rumble), something should be said. If not, why? The sentence at line 118 is not enough in my opinion and a more detailed presentation of the related biases should be provided. - There is no mention to the potential effect of evolving redox (e.g. when H2O+CH4 go to CO2 + H2) on the H2O $\delta$18O. Of course, the cap delta between H2O and minerals would not change, but the relative signatures would. This should be at least mentioned and/or justified. This is relevant because, for example, in the terrigenous layer, a fluid in equilibrium with graphite (not considered in the model) may be strongly enriched in one or the other C-bearing species relative to H2O. - Still on line 118: although the choice of considering molecular fluid species only does not fully reflect the technical means we have today (e.g. DEW model), I agree that this is probably the right choice for this early contribution. However, especially because this study centers on fluid-rock interactions and metasomatism, the

[Figure]

effect of omitting ionic species and their effect of potentially large mineralogical/mass changes has to be introduced. The manuscript cites a series of natural examples of strong fluid-mediated O resets. These rocks are in most cases associated with dramatic major element variations that cannot be explained without species more complex than molecular H2O. The possibility that these species have a negligible effect on the $\delta$18O of the system has to be demonstrated. For example, the capdeltabetweenHCO3-andH2OatroomTisabout40‰ÌĞthigherTitshouldbe lower, but maybe still significant if present in large amounts. At least for the carbonate layer, species like HCO3- may be important at the considered conditions (see Facq et al 2014 GCA). Here again I suggest providing more details on these assumptions and potential biases. See also the potential effect of pH on stable isotope variations (Ohmoto 1972). - F/R ratios. The only values of F/R ratios that I could find in the text (apologies if I am wrong) appear very low to me, especially in the case of channelized fluid flow. As time is present in the proposed model, it could help having some idea on how the proposed fluid/rock ratios translate into time-integrated fluid fluxes. The proposed values should at least in part correspond to the time-integrated fluid fluxes estimated in pervasive vs. channelized fluid systems in crustal settings (see review by Ague 2014 for example). F/R ratios alone do not provide insights on the hydrology of the systems and are sometimes meaningless (Baumgartner and Ferry 1991). I understand that many times this choice is imposed by the numerical code itself, but here you have the means to do this conversion at least once in the text, for reference. This could be also introduced at line 52. Line 19: bulk $\delta$18O value: in the source? Line 85: and also on the fluid speciation. . . that is not considered here but that can strongly modify the $\delta$18O evolution of the fluid/rock system. For example, at 500 âŮęC, the Cc-H2O and CC-CO2 cap delta for O differbyabout6‰ŁÌĞine110:"excluding"ismisleadinginmyopinion.Youmeanremov- ing from the reactive bulk, right? 161: can you clarify the meaning of natural profiles? Line 181-182: do the chosen values take into account processes like decarbonation? 193: this sounds like a model-driven assumption. Could you clarify? 323: $\delta$18O of the water: this is still a model assumption. I would say fluid instead. 344: increase in bulk $\delta$18O:

increase in the reactive bulk $\delta$18O? 345: reactive bulk $\delta$18O? 361-366: Here is where I miss the effect of decarbonation/dissolution and species other than H2O in the model. I suggest adding a sentence to recall the assumptions. 492: Airaghi et al: I suggest adding a couple more references on this topic.

---

## Author Comment (AC1) · 18 Dec 2019

Dear Editor, Dear Reviewer,

We appreciated the very detailed and constructive comments on different aspects of the study. We have addressed the suggestions given by the reviewer; in the following we report the point-to-point reply to the comments.

Best regards,
Alice Vho and co-authors

*At the date of review, the database, which is used for the modelling and which is fundamental for the models, is currently under review for another journal and can hence not be accessed. This is somewhat problematic regarding the final publication of this manuscript, at which point the database should be accessible to readers (see comment below).*
The manuscript underwent the last round of minor revisions and we are confident that it will be available soon. For the moment, the fractionation factors used for the calculations are also provided with the program (folder "WorkingDirectoy" of PTloop). This has been added in the section "Code availability". The file DO18db2.0.3.dat contains the fractionation factors between quartz and any end-member used in the calculation.

*10: For clarity, it should be added the Gibbs energy minimization is carried out for given whole-rock compositions.*
The text has been modified accordingly (line 11).

*15-20: The abstract would become clearer if a structure similar to the order/sections in the manuscript is adopted. For instance, the infiltration of an external fluid into mafic rocks is mentioned as point (1) in the abstract, but it is the last section in the results (3.3.4) and also near the end in the discussion. Better to first describe the simple case of dehydration only ("no interaction" case); it should also be mentioned that dehydration in the metasediments is also considered in the models. Second, the influx of MORB-derived fluid should be addressed; currently, MORB-derived fluids are not explicitly mentioned in the abstract, so this should be added as this process features prominently in the results and discussion. Finally, the influx of fluid derived from ultramafic rocks can be mentioned.*
Abstract modified accordingly. The term "MORB" has been replaced by the term "metabasalt" when indicating the subducting mafic oceanic crust. Metabasalt$_{(h)}$ refers to the hydrated MORB composition, while metabasalt$_{(a)}$ to the altered mafic oceanic crust (lines 163 – 165).

*82-90: Since sediment dehydration is also considered, this should be explicitly mentioned here. It is nicely described in the figure caption to Fig. 1 that the fluid produced in the "No Interaction" case is a mixture of MORB-derived and sediment-derived fluid.*
Two sentences have been added to clarify this point (lines 95 – 99).

*148: Reference to the database in Vho et al. (in review). As a reviewer of this manuscript for Solid Earth, I do not have access to this database and do not know the current status regarding publication of this database. This is somewhat problematic as the database is fundamental for the modelling performed in this manuscript. I am convinced that the authors are pursuing the publication of this accompanying manuscript as quickly and efficiently as possible, but I would like to see this the accompanying manuscript as accepted manuscript with doi number before the final version of this manuscript for Solid Earth goes online. The reason is that readers should be able to adequately follow and reconstruct all the information and proceedings of this article. This is difficult if the database is not accessible and may become problematic if the accompanying*

*manuscript is never published (I do not expect this to happen, but it seems a sensible approach notwithstanding the impeccable reputation of the present author team).*
See above.

*194: As these fractionation factors are explicitly mentioned, it would be useful to specify for which temperature the fractionation of 2 ‰ is representative of.*
It has been specified that this values if for T > 550 °C (line 209).

*203: white mica: The phase diagrams in Fig. 3 show "ph" (phengite), but in the text mostly "white mica" is used (but see line 212). Does "white mica" always stands for a potassic white mica? For consistency, only one term should be used throughout, and an explanation regarding the composition of the white mica would also be useful (e.g. the modelled composition might be a typical muscovite at lower grades but becomes more phengitic at higher temperatures and pressures).*
The composition of the modelled white mica has been added in the text. According to the composition, the term phengite was used, with the exception of the abstract where the term "white mica" (line 21) refers to all variety of potassic white mica compositions.

*212-224: There are a few minor issues in this section that should be addressed:*
*a) Modal phase changes are referred to partly in a neutral way (increase/reduction) but in other cases specific reactions are invoked (e.g. gln consumed in favour of jd+ank, lws breakdown producing grt). From a perspective of a metamorphic petrologist, one would be interested to see the full reaction equations. Two examples: If ankerite is produced, carbonate or CO2 is required as reactant; which other phases are involved in the lawsonite breakdown reaction? However, in the context of this study, this detail may not be necessary, and it may be suffice to formulate in a way without referring to specific reactions.*
This section was reformulated without referring to specific reactions since, as perceived also by the reviewer, this detail is not necessary.

*b) Clinopyroxene composition: As the change from omphacite to jadeite is mentioned, please clarify at which compositional boundary (mol% jadeite component) the change in name is made. Or is it a pure end-member jadeite? The coexistence of jadeite and omphacite should also briefly be addressed as in natural rocks, one would presumably expect only one clinopyroxene with changes in the jadeite-component in omphacite.*
Details on the pyroxene composition have been added in the text (lines 218 – 237). For consistency, also the composition of the garnet has been reported. A note about the interpretation of the coexistence of two pyroxenes has been added (lines 241 – 242) in line to what was already written for amphiboles.

*The authors point out that the co-existence of two amphiboles is of little relevance for the oxygen isotope modelling – is this similar for the pyroxenes? This should be clarified.*
Yes, it is similar for the pyroxenes and this point has been added in the text (lines 241 – 242).

*227: Initial water-saturated conditions: Please explain and justify the choice of water-saturated conditions, in particular for the fresh MORB. If one assumes that fresh MORB is initially composed of nominally anhydrous minerals only, where does the water come from? In the discussion later, the water released from the slab is dominated by the MORB-derived fluid, and presumably this is due to amounts of water stored under water-saturated conditions initially. Hence, does this initial assumption affect (at least some) of the model calculations, and how large is the effect? I appreciate that not all possible scenarios can be addressed in a single manuscript, but a brief justification of the choices made would be useful.*

In this case the main difference is that the amount of released fluid due to breakdown of hydrous phases would be smaller. This point has been added (lines 255-256).

*234: Glaucophane and actinolite and the intermediate fluid pulse: It seems in the figure that the growth of talc takes up the water released by consumption of actinolite and glaucophane, as the modal proportion of talc increases from 580 to 600°C, whereas the amount of water appears to increase at >600 °C when the modal proportions of talc and then lawsonite decrease. Please check carefully and modify the text accordingly.*

Talc takes only part (ca. 50%) of the water released by amphibole breakdown, because of the small abundance of talc (ca. 14 wt% in the metabasalt$_{(h)}$, ca. 6 wt% in the metabasalt$_{(a)}$, incorporating ca. 4 wt% of $H_2O$) with respect to the consumed amphiboles (ca. 50 wt% in the fresh MORB and ca. 40 wt% in the altered MORB, water content of $2.0 - 2.2$ wt%). This results in a peak of fluid release at 600 °C as can be observed also in figure 5. This point has been specified in the text (line 254).

*236: The liberation of water from the carbonate sediment is specified, but release of $CO_2$ is not mentioned. Does any release of $CO_2$ occur? Carbonate phases appear to remain stable, but the aspect should still be briefly explained for clarity.*

This is an important point and we acknowledge both the reviewers for the comments. The main reasons why we did not to include $CO_2$ in the calculation are the following. (1) The amount of $CO_2$ involved in this model is limited. The maximum $CO_2$ content in the altered MORB-derived fluid is ca. 10 mol% at 700 °C and 2.6 GPa, and it is lower ($2 - 7$ mol%) at the conditions of the major fluid pulses. In the carbonate sediment the $CO_2$ content is $> 7$ mol% at T $> 560$ °C and P $>$ 2.18 GPa, where a negligible amount of fluid (i.e. $<< 0.01$ vol%) is released. (2) The oxygen isotope fractionation between $CO_2$ and $H_2O$ is still poorly constrained and presents limitations (see below); therefore it was not included in the internally consistent database version used for this study. Published experimental calibrations involving $CO_2$ are limited (e.g., Böttcher, 1994 for norsetite-$CO_2$; O'Neil and Adami, 1969 for $H_2O$-$CO_2$) and were performed at T $< 100$ °C, making the validity of the available fractionation factors at high temperature questionable. Zheng (1994) provides fractionation data for calcite-$CO_2$, quartz-$CO_2$ and $H_2O$-$CO_2$; however, incremental calculations have strong limitations and must be used with caution (e.g., Chacko et al., 2001). For the $H_2O$-$CO_2$ pair, the available calibrations (Friedman and O'Neil, 1977; O'Neil and Adami, 1969; Zheng, 1994) are in strong disagreement and predict fractionations of 1.50 ‰, -1.87 ‰ and -4.41 ‰ at 700 °C and of -8.85 ‰, -11.45 ‰ and -10.99 ‰ at 350°C respectively. Overall, the $H_2O$-$CO_2$ fractionation is large (-5 $-$ -12 ‰) at T $< 440$ °C, where $CO_2$ is absent or present in negligible amount in the fluid phase in our model; it decreases to -2 $-$ -6 ‰ at T = 550 °C, where the amount of $CO_2$ present in the fluid phase in our model is minor (ca. 3 mol% in the fluid released by the MORB, ca. 6 mol% in the fluid released by the sediment); it is moderate (ca. -5 ‰) to absent (depending on the chosen calibration) at T $\geq 600$ °C, where the amount of $CO_2$ in the fluid increases. The consideration of the $CO_2$ component would produce a negligible to minor shift on the fluid $\delta^{18}O$ at the condition of significant release ($0.1 - 0.2$ ‰ at 520 °C and $0.0 - 0.6$ ‰ at 620 °C for the MORB-derived fluid and $0.1 - 0.3$ ‰ at 480 °C and $0.2 - 1.1$ ‰ at 620 °C for the sediment-derived fluid, depending on the calibration).

We added a paragraph in the section 3.2 describing the potential effect of the $CO_2$ component present in the fluid released by the altered MORB and the carbonate. The title of the section has been also changed from "Production of water" to "Production of aqueous fluid". However, we did not introduce $CO_2$ in the computation for the reasons discussed above.

*282: Mafic fluid (see also 288, 297, 300 and elsewhere): Using the terms mafic fluid and ultramafic fluid is not appropriate and should be avoided. The term "mafic" is derived from magnesium and ferrum (iron) rich, which is appropriate for rock compositions but not for the*

*fluids considered here. The same applies to "ultramafic fluids" (e.g. in lines 309, 311 and elsewhere), serpentinite-derived fluid should be used instead.*

The term "mafic" has been replaced by "metabasalt-derived" and term "ultramafic" by "serpentinite-derived" when referring to fluids.

*311: This statement is a bit vague. What exactly is the effect in the PI and NI cases on MORB? If the variations in the sedimentary rocks decrease to zero, does that mean there is no effect at all, or no change compared to the previous cases? Please formulate more precisely here.*

The sentence has been rephrased and moved to the end of the section in order to make the point clearer.

*331-337: The example of the granite appears to be out of place here, as granite has not been considered anywhere else in the manuscript. A dry basalt would be a more appropriate example, which can be linked to the scenarios considered much better. But the results presented show the limited effect on the O isotope variation anyway, so consider deleting this section altogether.*

This example has been moved to the supplementary material S4.

*363-364: This statement is important, and could be highlighted in abstract and/or conclusions.*

The statement has been added in the abstract (line 17) and in the conclusions (lines 576 – 578).

*387-393: This section would benefit from a few more details regarding the studies on oxygen isotope zoning in metamorphic minerals, and how the modelling results can be linked to these results (and possibly used to support interpretations or argue for alternative interpretations). Questions that are of interest to the reader include: What kind of zonation was observed in the minerals studied? With which of the modelled scenarios do these patterns coincide? Providing more details here and some specific examples would also be useful to emphasize the wider implications for studies based on natural samples.*

Three examples of observed intragrain $\delta^{18}O$ variations in garnet from HP rocks have been reported (Martin et al., 2014; Rubatto and Angiboust, 2015; Vielzeuf et al., 2005b; lines 431 – 434).

*403: Integrated Fluid/rock ratios: It is not entirely clear where the numbers come from as they have not been mentioned before. Please clarify.*

The concept of integrated fluid/rock ratio has been now defined in the section 2.1 as "as the total mass of aqueous fluid that has passed through and interacted with the rock normalized to the mass of the rock".

*410: Serpentinite-derived fluid input into the sedimentary layer: Is this fluid in the models not a mixture of serpentinite-derived fluid and MORB-derived fluid since MORB also dehydrates? If so, clarify this point.*

Yes, it is a mixture and the point has been clarified (lines 453 – 454).

*413-415: Detection of serpentinite-derived fluids: It would be useful if the authors could refer to the (possible) detection of such fluids in real sediments to underline the relevance of their study.*

Two natural examples have been reported (lines 457 – 462). Martin et al. (2014) describe a shift in $\delta^{18}O$ of -2.5 ‰ among different generations of HP garnet in a sample from the Corsica continental basement (garnet mantle $\delta^{18}O$ = 7.2 ± 0.4 ‰, garnet rim $\delta^{18}O$ = 4.7 ± 0.5 ‰). The authors associate this shift to an infiltration of serpentinite-derived fluids and, to a lesser extent, altered gabbro-derived fluid. Williams (2019) describe an extreme $\delta^{18}O$ shift of -15 ‰ between garnet core and rim in a metasediment from the Lago di Cignana Unit. Such an oxygen isotope composition variation has been related to a strongly channelized fluid influx originated from the dehydration of serpentinites.

*416-418: Effects: The relatively "dry" system still starts with a water-saturated MORB; so the reader may wonder how things change if the system is really dry – would the oxygen isotopic effects even larger? (see also earlier comments).*

The terms "wet" and "dry" have been removed for clarity: relatively water-rich and relatively water-poor systems are used instead. The main difference with considering undersaturated basalt as starting composition would be the release of less fluid (as has been specified at lines 155-256) and therefore less capacity to infiltrate upper lithologies and modify their $\delta^{18}O$ value.

*419-437: This section seems rather unnecessary because it does not add much to the discussion on oxygen isotopes, the main statement emphasizing that the trends are similar to the ones shown earlier. The discussion on water release is fine but key points could be incorporated into the section "Model geometry" where some of the differences between the P-T paths are already highlighted.*

We believe that this section is important to provide an overview of the possible variations associated to the use of different P-T paths for the model. It also serves the purpose to clarify the doubt on whether the chosen geotherm is representative for any natural system (see the comment on figure 2 below). Therefore, it has been kept in the text.

*450-468: Can the relevant equations that consider the subduction rate and the volumes of fluid released be shown here so that readers get a better understanding of the modelling approach?*

A clarification about how the chosen subduction rate controls the amount of water infiltrating at a given point of the slab mantle interface has been added (lines 509 – 512). Given the column length of 1 m, a subduction rate of 1 cm/y implies that any fixed point (i.e. fixed P-T conditions) at the slab/mantle interface receives in 100 years the total amount of fluid that a single column can liberate at those conditions. Hence, in this example 4892.6 kg of water/100 years (i.e. the amount released by the considered column at the chosen conditions, as explained in the text) infiltrate the mantle wedge.

*475: What exactly are "high" $\delta^{18}O$ arc lavas. Please provide some values or a range of values.*

Values given in the cited studies have been reported (phenocrysts in lavas from Central Kamchatka: olivine $\delta^{18}O$ = 5.8 – 7.1 ‰ and clinopyroxenes $\delta^{18}O$ = 6.2 – 7.5 ‰, Dorendorf et al., 2000; New Guinea: silicate glass inclusions in olivine $\delta^{18}O$ = 8.8 – 12.2 ‰, clinopyroxenes in metasomatized lehrzolite $\delta^{18}O$ = 6.2 – 10.3 ‰, Eiler et al., 1998).

*486: Another important statement relevant for the interpretation of natural samples, which may also be emphasized in the conclusions.*

This statement has been added in the conclusions (lines 597 – 599).

*515: Interesting aspect which may be of interest to studies on natural serpentinites. For instance, have such elevated O isotope signatures been observed in natural wedge serpentinites? Or can O isotopes be used to distinguish wedge from abyssal peridotites in the geological record? Briefly expanding on these aspects would widen the relevance of this study.*

Theoretically, mantle wedge metasomatized/serpentinized rocks after interaction with slab-derived fluids are expected to increase the $\delta^{18}O$ with respect to the mantle signature of 5.5 ‰. It appears however impossibile to use this point as main discriminant to distinguish sea-floor (or in general low-T) serpentinites from HP ones because the first type is highly variable in oxygen isotope composition (ranging between 1 and more than 10 ‰).

*Figure 1: As in the text, please avoid the terms "mafic fluid" and "ultramafic fluid". Add $\delta^{18}O$ to the numbers given in the figure. Clarify that 4.5 ‰ is a fluid value, not the value of the serpentinite. Give the sources for the $\delta^{18}O$ values used in the figure, or refer to the text.*
Figure and figure caption modified accordingly. To avoid confusion, only the $\delta^{18}O$ of the rocks (fluid sources) has been reported in the figure.

*Figure 2: The meaning of the abbreviation D80 should be explained. Moreover, one may wonder whether an average geotherm is useful as it may not represent any real subduction zone. Typo in the figure "Syracuse". Regarding the lines in this figure and in other figures, they are dashed rather than dotted and should be labelled accordingly.*
The meaning of D80 has been explained as "the geotherm dominated by a steep T gradient at 80 km depth, which occurs at the transition from partial to full coupling". The implications of the choice of a specific geotherm for the model, and the possible variations occurring when the P-T path is modified, are discussed in the section 4.4 "Effect of the subduction geotherm".
The typo has been corrected and "dashed" has been used for the lines.

*Figure 3: Mineral abbreviations should be explained. It would also help to indicate initial water contents in the figure or the caption.*
Mineral abbreviation reference to Whitney and Evans (2010) has been added. Titanite field colour has been changed to be consistent with Fig. 4. The initial water content in vol% (< 1 vol% for the MORBs and the carbonate sediment, ca. 3 vol% for the terrigenous sediment) have been added in the caption together with a reference to Table 1 for details.

*Figure 4: The line for titanite is almost invisible in a print out, a somewhat darker colour would improve visibility. For diagrams (g) and (h), I recommend presenting separate diagrams for the partial and high interaction cases because the distinction of the lines marked with stars is not very clear (the bulk trend could be copied into the respective other diagram for comparison). As above, lines are dashed (short bars) rather than dotted (points).*
The colour of the line for titanite has been modified. The term "dashed" has been used for the lines. The figure has been split into two figures in order to make the diagrams (g) and (h) clearer. Figure 4 includes the diagrams (a), (b), (c), (d), (e) and (f) of the original figure, while figure 5 include 4 diagrams showing separately the partial interaction and the high interaction cases for both the sediments. In all the diagrams we plotted the no interaction case lines for comparison. Figure captions and references in the text have been modified accordingly.

*Figure 6: Avoid terms "mafic" and "ultramafic" fluid.*

Figure and figure caption modified.

*Technical corrections:*

All addressed.

---

## Author Comment (AC2) · 18 Dec 2019

Dear Editor, Dear Reviewer,

We appreciated the constructive and helpful comments that shed light on important details that were missing in the previous version of the manuscript. We have addressed the suggestions given by the reviewer and the point-by-point response is reported in the following.

Best regards,
Alice Vho and co-authors

*Chemical system: carbon is not present in the list at line 164.*
It has been specified which lithologies contain carbon (line 177).

*It is not clear if CO2 (as well as other species such as CH4 and H2) in the fluid was considered or forced not to form. It is clear, however, that CO2 and other species (if any) were not considered in the 18O budgets (line 118). Same for S. A sentence should be added.*
We acknowledge the comments of both the reviewers about the importance of C since it is present both in the altered MORB and in the carbonate sediment. A sentence has been added at the end of the section 2.2 (line 118 in the original manuscript) that refers to the newly added discussion about the amount and possible effects of $CO_2$ in our model (section 3.2, see also the answer to the comments from reviewer 1). The implementation in the model of other species (i.e. $CO_2$ as well as $CH_4$, $H_2$) could be assessed, provided that (1) reliable constraints on the oxygen isotope fractionation between these species and water or minerals are determined and (2) their consistency with other available data is established. However, $CH_4$ and H2 do not contain any oxygen, being less relevant for the model than $CO_2$. This point has been added in the section 4.6 "Model applications and future directions".
S is not present in any of the considered bulk compositions and therefore no S species are involved in this model. Moreover, oxygen isotope fractionation between water and S-species is poorly constrained, especially at T > 350 °C, where no data are available.

*The text should clarify if CO2 was considered as a negligible parameter in this model (non just not considered). To be honest, I do not see how percolation of potentially high fluid fluxes through the carbonate layer should not mobilize (not just equilibrate) a large portion of the bulk carbonate O. Take the example of Ague and Nicolescu (2013 Nat Geo): an almost complete carbonate devolatilization along a fluid channel. Or the reverse carbonation (Piccoli et al 2016; Scambelluri et al 2016). Can O-bearing fluid species other than H2O modify the model assumptions? If yes (e.g. Baumgartner and Rumble), something should be said. If not, why? The sentence at line 118 is not enough in my opinion and a more detailed presentation of the related biases should be provided.*
A discussion on the possible effects of $CO_2$ in our model have been added in section 3.2 (see above and answer to reviewer 1). The anticipation of future directions that might consider decarbonation/carbonation reaction, or more in general mineral dissolution, transfer and re-precipitation has been added in the section 4.6 "Model applications and future directions".

*There is no mention to the potential effect of evolving redox (e.g. when H2O+CH4 go to CO2 + H2) on the H2O δ18O. Of course, the cap delta between H2O and minerals would not change, but the relative signatures would. This should be at least mentioned and/or justified. This is relevant because, for example, in the terrigenous layer, a fluid in equilibrium with graphite (not considered in the model) may be strongly enriched in one or the other C-bearing species relative to H2O.*

This effect of oxidation state on C, S and Cl stable isotope partitioning has been described (e.g., Chacko et al., 2001; Sharp, 2017). As explained by Sharp (2017), oxygen has one oxidation state and so it is not affected by the redox changes that occur in most of the other elements used for stable isotope studies. The heavy isotope of oxygen will be preferentially fractionated into short, strong chemical bonds (such as $Si^{4+}$) generally with a high oxidation state, however this is not always the case (for example, uraninite $U^{4+}O_2$ strongly incorporates $^{16}O$ relative to quartz), so that oxidation state alone does not always correlate with oxygen isotope enrichment. Therefore no evolving redox effect has been considered.

*Still on line 118: although the choice of considering molecular fluid species only does not fully reflect the technical means we have today (e.g. DEW model), I agree that this is probably the right choice for this early contribution. However, especially because this study centers on fluid-rock interactions and metasomatism, the effect of omitting ionic species and their effect of potentially large mineralogical/mass changes has to be introduced. The manuscript cites a series of natural examples of strong fluid-mediated O resets. These rocks are in most cases associated with dramatic major element variations that cannot be explained without species more complex than molecular H2O. The possibility that these species have a negligible effect on the δ18O of the system has to be demonstrated. For example, the capdelta between HCO3- and H2O at room T is about 40‰. At higher T it should be lower, but maybe still significant if present in large amounts. At least for the carbonate layer, species like HCO3- may be important at the considered conditions (see Facq et al 2014 GCA). Here again I suggest providing more details on these assumptions and potential biases.*

Oxygen isotope partitioning between $HCO_3^-$ and $H_2O$ (as well as for other dissolved C-species) are poorly constrained and the data are obtained at low T (i.e. Halas and Wolacewicz, 1982, 25 – 45 °C; Usdowski and Hoefs, 1993, 19 – 25 °C). Therefore, any extrapolation to the temperature range relevant for this model and discussion on possible effects on the $\delta^{18}O$ partitioning among phases is disputable.

The study of Facq et al. (2014) points out the importance of $HCO_3^-$ and $CO_3^{2-}$ based on experiments on a very special system (a single aragonite crystal in water). They conclude that ion-pairing in deep crustal and mantle aqueous fluids may occur during the dissolution of carbonate minerals at high pressure, even if in natural system the complex interplay of pressure, temperature, and activity ratios imposed by the silicate and/or carbonate environment must be considered. Even if we would consider the presence of these C-species instead of $CO_2$ at high pressure, and we would assume to be able to extrapolate up to 700 °C the low T experimental data for oxygen isotope fractionation among them and $H_2O$ (Usdowski and Hoefs, 1993), the fractionation between $HCO_3^-$ / $CO_3^{2-}$ and $H_2O$ is smaller than the one between $CO_2$ and $H_2O$, resulting in an even smaller effect than the one discussed for $CO_2$ in section 3.2. We acknowledge the importance of this study, but we believe that this is a very specific point still under investigation and there are too many uncertainties in the available data to consider it at this stage of the model. However, we mention the possibility of introducing additional C-species in future developments in the section 4.6 "Model applications and future directions".

*See also the potential effect of pH on stable isotope variations (Ohmoto 1972).*

Ohmoto (1972) described the effect of pH on S and C stable isotopes. The effect of pH state on C and S stable isotope partitioning has been described also in the more recent studies (Chacko et al., 2001; Sharp, 2017). No major pH effect on O stable isotope partitioning has been recognized with the exception of Fe(III)-oxides, for which large variations in experimental results at T < 40 °C might be attributed also – but not exclusively – to the difference in pH (Chacko et al., 2001).

*F/R ratios. The only values of F/R ratios that I could find in the text (apologies if I am wrong) appear very low to me, especially in the case of channelized fluid flow. As time is present in the proposed model, it could help having some idea on how the proposed fluid/rock ratios translate*

*into time-integrated fluid fluxes. The proposed values should at least in part correspond to the time-integrated fluid fluxes estimated in pervasive vs. channelized fluid systems in crustal settings (see review by Ague 2014 for example). F/R ratios alone do not provide insights on the hydrology of the systems and are sometimes meaningless (Baumgartner and Ferry 1991). I understand that many times this choice is imposed by the numerical code itself, but here you have the means to do this conversion at least once in the text, for reference. This could be also introduced at line 52.*

Values for integrated fluid/rock ratios (as defined in section 2.1, lines 88 – 89) in the sediments have been added in the results (section 3.3.1). In case of high interaction, the integrated F/R ratios are 0.75 kg/kg in the carbonate sediment (corresponding to 2.1 F/R volume ratio) and 0.35 kg/kg in the terrigenous sediment (corresponding to 0.98 F/R volume ratio). They drop to 1/2 in case of partial interaction. These values are consistent with the F/R ratios calculated by Konrad-Schmolke et al. (2011) of 0.15 – 0.3 for weakly deformed samples and 0.5 – 4 for mylonites in the Sesia Zone. Ague (2014) calculates fluid fluxes in the order of 1000 $m^3/m^2$ (and up to $10^4$ – $10^5$ $m^3/m^2$ in case of channelized fluid flow) for crustal column of 15 km. Our crust is 1 km thick and the fluid fluxes are 160 – 170 $m^3/m^2$, therefore comparable in the order of magnitude with the one calculated by Ague (2014).

*Line 19: bulk δ18O value: in the source?*
Yes. It has been specified (line 17).

*Line 85: and also on the fluid speciation that is not considered here but that can strongly modify the δ18O evolution of the fluid/rock system. For example, at 500 °C the Cc-H2O and CC-CO2 cap delta for O differ by about 6‰.*
A discussion about the effect of a mixed $H_2O$-$CO_2$ fluid on the $\delta^{18}O$ has been added in the section 3.2 "Production of aqueous fluid" (see above).

*Line110:"excluding" is misleading in my opinion. You mean removing from the reactive bulk, right?*
The term "removing" has been used instead of "excluding".

*161: can you clarify the meaning of natural profiles?*
The sentence has been modified for clarity (line 173).

*Line 181-182: do the chosen values take into account processes like decarbonation?*
Those are the starting $\delta^{18}O$ values (25 – 35 ‰, retrieved from marine sediment measurements, where no decarbonation occurred). Possible decarbonation during subduction might decrease the starting $\delta^{18}O$ (i.e. because calcium carbonate has usually higher $\delta^{18}O$ than the bulk $\delta^{18}O$), and indeed the $\delta^{18}O$ of carbonate in HP metamorphic terrains could be lower. However, as already mentioned, decarbonation has not been considered at this stage of the investigation, but represents an important, although challenging, development as has been stated in the section 4.6.

*193: this sounds like a model-driven assumption. Could you clarify?*
The sentence has been modified (lines 206 – 208) to clarify that the choice of the serpentine $\delta^{18}O$ = 2.5 ‰ has been done in order to maximize the difference in $\delta^{18}O$ between the fluid-source and the fluid-sink lithologies, while using a feasible value for natural serpentinites. Any interaction with higher- $\delta^{18}O$ serpentinite-derived fluids will just reduce the effects described in this study.

*323: δ18O of the water: this is still a model assumption. I would say fluid instead.*
The text has been modified accordingly (line 357).

*344: increase in bulk δ18O: increase in the reactive bulk δ18O?*

Yes, the text has been modified accordingly (line 373).
.

*345: reactive bulk δ18O?*
Yes, the text has been modified accordingly(line 373).

*361-366: Here is where I miss the effect of decarbonation/dissolution and species other than H2O in the model. I suggest adding a sentence to recall the assumptions.*
The assumption has been recalled (line 393).

*492: Airaghi et al: I suggest adding a couple more references on this topic.*
Few more references have been added (Cartwright and Barnicoat, 2003; Engi et al., 2018; Konrad-Schmolke et al., 2011; Rubatto and Angiboust, 2015) (line 549).